# The impacts of fine-tuning, phylogenetic distance, and sample size on big-data bioacoustics

**Kaiya L. Provost**[ORCID]*, **Jiaying Yang, Bryan C. Carstens**

Department of Evolution, Ecology and Organismal Biology, The Ohio State University, Columbus, Ohio, United States of America

* provost.27@osu.edu

## Abstract

Vocalizations in animals, particularly birds, are critically important behaviors that influence their reproductive fitness. While recordings of bioacoustic data have been captured and stored in collections for decades, the automated extraction of data from these recordings has only recently been facilitated by artificial intelligence methods. These have yet to be evaluated with respect to accuracy of different automation strategies and features. Here, we use a recently published machine learning framework to extract syllables from ten bird species ranging in their phylogenetic relatedness from 1 to 85 million years, to compare how phylogenetic relatedness influences accuracy. We also evaluate the utility of applying trained models to novel species. Our results indicate that model performance is best on conspecifics, with accuracy progressively decreasing as phylogenetic distance increases between taxa. However, we also find that the application of models trained on multiple distantly related species can improve the overall accuracy to levels near that of training and analyzing a model on the same species. When planning big-data bioacoustics studies, care must be taken in sample design to maximize sample size and minimize human labor without sacrificing accuracy.

**Data Availability Statement:** Raw recordings are available publically from the Borror Lab of Bioacoustics (blb.osu.edu) and Xeno-Canto (xeno-canto.org). Code is available on KLP's github page github.com/kaiyaprovost/bioacoustics. All other

## Introduction

### Avian bioacoustics

Vocalization is an important form of communication across many taxa. Birds (class Aves) make and respond to vocalizations profusely, for example in territory defense [1], mate choice [2–4], both conspecific and heterospecific communication [5, 6], species discrimination [7, 8], prey detection [9], and food solicitation via begging calls [10, 11]. Bird vocalizations are generally classified into songs and calls. Significant attention has been paid to song vocalizations, which are thought to mediate sexual selection [12–14] and be important in the generation of reproductive isolation in many bird taxa [15–18]. Though less prominent, bird calls are also important as they can mediate non-sexual behaviors [19], for instance between parents and offspring [20] or even intra-specific communication [21]. Despite evidence that both the biotic

data are available here: https://doi.org/10.5061/dryad.8pk0p2nrb.

**Funding:** KLP and BCC were supported by the National Science Foundation DEB-2016189 [https://www.nsf.gov/awardsearch/showAward?AWD_ID=2016189]. The funders had no role in study design, data collection and analysis, decision to publish, or preparation of the manuscript.

**Competing interests:** The authors have declared that no competing interests exist.

and abiotic environment can impact the development and evolution of vocalization, there are few studies that examine how the community soundscape evolves in tandem (but see [22, 23]). Fortunately, an abundance of bioacoustic data is available.

The sheer volume of recording data available to modern researchers represents a tremendous asset to avian bioacoustic research. For example, the citizen-science initiative Xeno-Canto (xeno-canto.org) having ~700,000 recordings and the Macaulay Library bioacoustics repository (which includes eBird, [24]) having ~1.1 million recordings (birds.cornell.edu/MacaulayLibrary). Even the smaller bioacoustic collections, which represent important repositories of historical recordings, can have tens of thousands of songs to parse (e.g., the Borror Laboratory of Bioacoustics with ~36,000, blb.osu.edu). This volume of data represents a methodological challenge because most bioacoustic workflows require the songs to be segmented [11]. This segmentation necessitates identifying sounds of the target species from noisy backgrounds (e.g., recordings taken in windy places or areas with anthropogenic noise), from non-target species, or, for more complex analyses, identifying individual syllables of song [e.g., 25–29]. To date, this process still requires careful tuning from researchers. Ideally, automated methods would be useful if they could reduce the amount of human interaction without compromising data quality, as song segmentation is extremely time consuming (e.g., [30] found manual annotation to take nearly five times as long, even after accounting for needing to check automatic annotations), though less so than the gold-standard methods of performing playback experiments to directly evaluate behavioral responses to different vocalization treatments (e.g., [31] performed ~4,600 minutes of playback experiments to investigate a single species). Automatic segmentation has been performed in a multitude of ways, with a proliferation of methods that use heuristics to detect sound onsets and offsets. These heuristics can include noise thresholds, similarity to previously identified segments, amplitude, or frequency [32]. While these methods are highly effective at segmenting syllables for single species, re-parameterizing them to work on multiple species (and species with very diverse syllable types) is challenging. Here we ask whether we can automate this process on multiple species simultaneously using deep learning models, and importantly whether we can leverage existing large publicly available datasets to minimize the need for excessive supervision by researchers.

## Machine learning

Machine learning, which can be either unsupervised or supervised, is a useful tool for data processing. Unsupervised machine learning is typically used to cluster data [33–35]. In contrast, supervised learning is frequently used to make predictions [36]. Many well-known algorithms in ecology and evolution are derived from machine learning models in some capacity, for example k-means clustering [37–39]. However, most of these techniques require inputs that are pre-defined, such that the presence or absence of features and variables must be determined beforehand. This can be difficult when working with data types that are highly continuous, highly variable, poorly understood, or difficult to convert into quantitative numerical metrics.

More recently, deep learning via artificial neural networks (ANNs) has become popular [40]. ANNs are made from many layers of nodes which construct weighted mathematical functions to perform tasks, typically from input data with diagnostic features [41]. Deep learning methods are methods where the features used do not need to be prepared before using them for training. Because ANNs can iteratively learn their own features, use large amounts of data, and are less constrained by assumptions about that data, they are extremely flexible and can handle many kinds of tasks [e.g., 41–47]. These methods have been used on a variety of topics including image processing, video segmentation, and speech recognition [48–50].

Although challenges remain with respect to scalability, computational efficiency, and how to handle depauperate data [47], deep learning is one of the most powerful analytical tools in the modern researcher's toolbox, particularly when human knowledge is lacking, or datasets are too large to be workable by traditional means.

In the context of ecology and evolutionary biology, there have been many recent applications of both shallow and deep machine learning, including population genetics and phylogeography [e.g., 51, 52], bioacoustics [e.g., 53–55], species classification [e.g., 56], phylogenetics [e.g., 57, 58], sequencing and genomics [e.g., 59, 60], and phenotypic analyses and morphometrics [e.g., 61, 62]. Neural networks and support vector machines tend to be the most applied algorithms in these analyses. However, the development of new machine learning techniques is rapid-pace, and evaluations of these methods on empirical data are often missing, particularly when it comes to the development of best-practices (but see [63]).

## TweetyNet for song segmentation

Though there are many ways to categorize machine learning algorithms, a useful one in biology is to categorize evaluative vs extractive algorithms. Evaluative algorithms are typically used to make predictions from data, for example by training a model to distinguish between simulated evolutionary scenarios [51, 52, 64–66]. Extractive algorithms, on the other hand, are designed to process data in some way. With respect to bioacoustics, an extractive algorithm could, for example, segment out syllables within vocalizations. A recently developed application, TweetyNet, was released to perform just this task [55] using deep learning via ANNs. Specifically, TweetyNet uses convolutional and recurrent ANNs. This model has been shown to be highly accurate on complicated syllable types (i.e., canary songs, [55]) even with as little as 180 seconds of training data. Further, the algorithm can segment not only syllables from background noise but classify the syllables into user-determined syllable types. As such, TweetyNet is one extractive algorithm that could be useful in reducing the human workload in bioacoustics.

Despite these advances, there is at present no evaluation of this performance on multiple species at once. Though potentially much faster than hand-processing data, TweetyNet and ANNs like it can still take prohibitively long times to train. Therefore, minimizing training time can be accomplished by co-opting existing trained ANNs to perform similar tasks, known as transfer learning. This can be used to take a neural network trained on one batch of data, and then applying it to new, related data with relatively little further training [47, 67]. One well-known example outside of bioacoustics is that of bidirectional encoder representations from transformers, or BERT [68]. This is a massive model trained on ~3.3 billion words to perform two tasks: 1) predicting a word within a sentence, and 2) predicting whether two sentences follow each other. In learning these tasks, BERT developed an understanding of how the English language works, and as such numerous applications can be added onto BERT to perform similar language processing tasks (e.g., word suggestions while typing). Importantly, this is without having to re-train the entire BERT model. Other later-generation models that are used similarly for language processing include the GPT family of models [69] and T5 models [70].

With respect to bioacoustics, it is not currently known how well models could generalize (via transfer learning and fine-tuning) when looking across multiple different species that it has not explicitly been trained on. Fine-tuning a more general model to fit a specific case, like the BERT example, would necessitate building one massive model that incorporates many examples [71–73]. For example, if a model exists that can segment all existing organismal sounds, a researcher could modify that model to segment only one species of bird without

having to completely re-train the model. However, there is not a good understanding at present as to whether a model trained on a specific task can work on a different, but similar, specific task. In the bioacoustics example, this would be if a model exists that can segment Species A, which is then applied to segment Species B without any downstream modifications whatsoever.

Here we compare different treatments of bioacoustic data to optimize the overall performance of machine learning models trained to segment syllables of avian song. We investigate the utility of these methods across a subset of bird species, including birds from different clades ranging in divergence times up to ~85 million years. We use those taxa to evaluate how well models perform on closely vs distantly related taxa for use in maximizing computational efficiency. Though the evolution of song is rapid in part due to the roles of cultural evolution and sexual selection [11], sound production is also phylogenetically constrained [74, 75]. As such, we hypothesize that taxa that are more distantly related should have less similar songs, and therefore models should perform worse with increasing species divergence. Finally, we provide suggested practices for moving forward with neural networks in avian bioacoustics. Our intent is to guide researchers working in bioacoustics to know how to fine-tune existing machine learning algorithms, as well as to put forth a fully trained TweetyNet model for future use by researchers.

## Materials and methods

Unless otherwise noted, all analyses were performed using a MacBook Pro running macOS Catalina (10.15.7) with a 2.8 GHz i7 core and 16 GB of RAM. Data is archived on Dryad (https://doi.org/10.5061/dryad.8pk0p2nrb). Code is available on GitHub at github.com/kaiya-provost/bioacoustics.

### Acquiring and filtering song data

We downloaded bioacoustics data from two song databases: Xeno-Canto (xeno-canto.org), and the Borror Laboratory of Bioacoustics (blb.osu.edu). We used the Xeno-Canto database as it has a large amount of data of varying quality, due to it being a citizen science initiative, and a convenient R package exists for accessing and filtering the files. The Borror Lab of Bioacoustics database was used because it is full of historical data and based in North America, the region we wished to focus on, in addition to being the lab where the authors are based. In both the Borror Lab of Bioacoustics and the Xeno-Canto database, many species have a small number of recordings. For the former, nearly 9,400 bird species lack any data, with an additional ~400 species with fewer than 10 recordings. For the latter, ~600 species lack any data, with an additional ~2,500 with fewer than 10 recordings (Fig 1). As such, having methods which can handle species with small numbers of recordings is critical.

We targeted songs from *Cardinalis sinuatus*, *Cardinalis cardinalis*, *Passerina amoena*, *Zonotrichia leucophrys*, *Poecile carolinensis*, *Vireo altiloquus*, *Myiarchus tuberculifer*, *Empidonax virescens*, and *Calypte anna*. We chose these taxa because they were distributed across the phylogenetic tree, had relatively high numbers of recordings available, and most of the species had previously estimated song variation and complexity measures [76] for downstream analyses. We also targeted songs from *Melozone fusca* which we used to evaluate the performance on a novel species. This selection of ten taxa covers two bird orders (~70% of bird species) including the single largest order, six bird families (~10% of bird species), and nine genera (~1% of bird species). Furthermore, it encompasses two of the three main bird clades that have vocal learning, Trochilidae (*Calypte*) and Passeri (most other species). It also samples the sister taxon to the latter, Tyranni (*Myiarchus* and *Empidonax*), which are not vocal learners. This

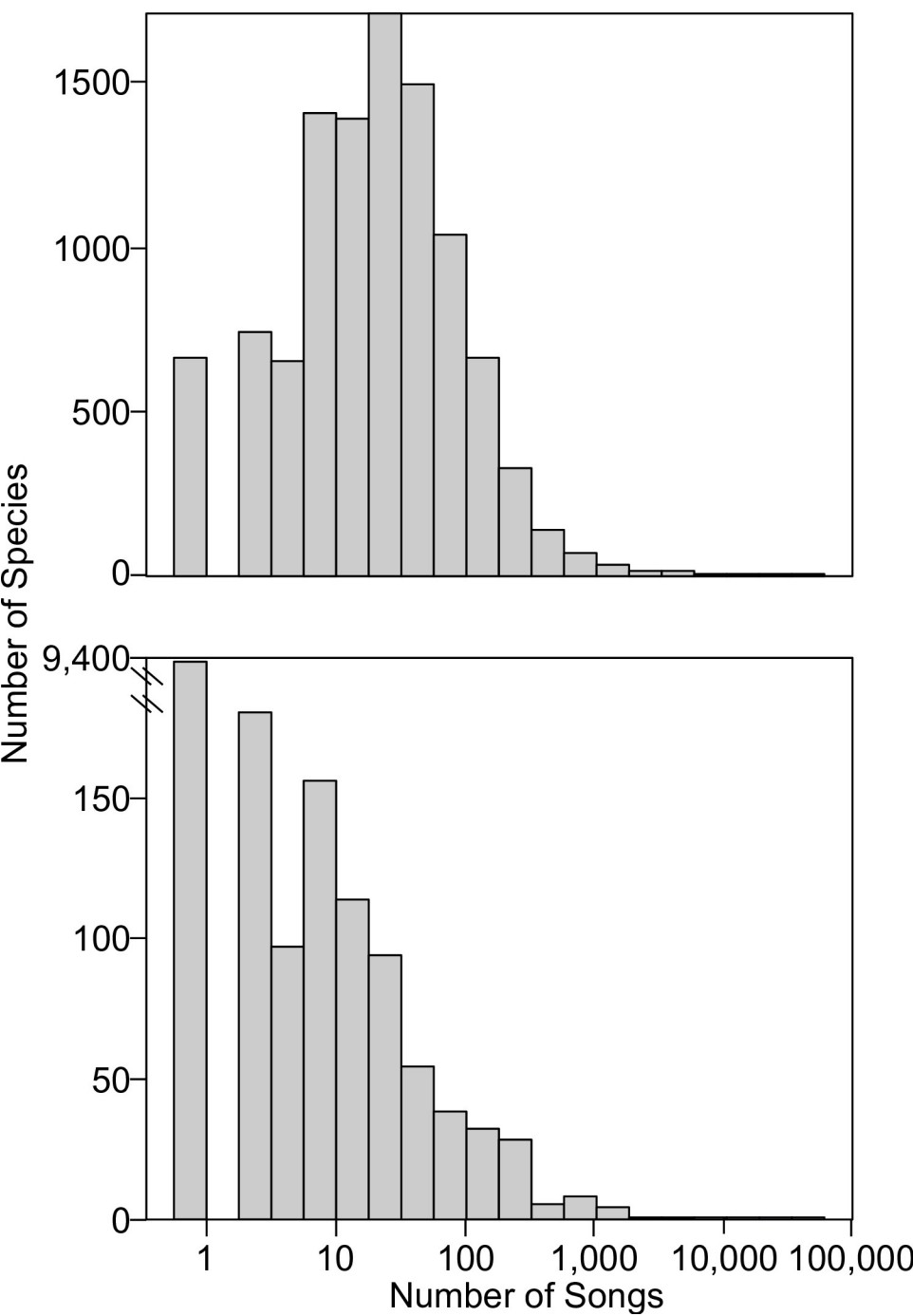

**Fig 1. Most species of bird have little bioacoustic data.** X-axis shows, on a log scale, the number of songs present per species. Y-axis shows the number of species per song bin. Top panel: Xeno-Canto data. Bottom panel: Borror Laboratory of Bioacoustics (BLB) data; note the x-axis has a gap to show an exceedingly high number of species with no data, as BLB is North America focused.

taxonomic, phylogenetic, and behavioral diversity lends itself well to general conclusions across these models.

For the Xeno-Canto songs, we restricted the data downloaded to all those in North America that were of Quality "A" according to the Xeno-Canto database using the package warbleR

version 1.1.26 [77] in R version 4.0.3 [78]. Xeno-Canto recordings are available in the MP3 format, but since our downstream pipeline requires WAV format, we converted them using the package tuneR version 1.3.3 [79]. For the Borror Laboratory of Bioacoustics songs, we downloaded all available data from all species, which were already available as WAV format. Downstream analyses required recordings of the same sample rate and will not work on stereo recordings. As such, stereo recordings were converted to mono using the 'mono' function in tuneR. We also converted recordings to a sample rate of 48,000 Hz, which was the most common sample rate in our dataset, using the 'resamp' function in seewave version 2.2.0 [80].

## Identification of syllables via manual and automatic annotation

We arbitrarily chose a subset of recordings to annotate from each species. As few as three minutes of recording is sufficient for the TweetyNet algorithm to perform accurately [55], so we ensured that all species had at least 180 seconds worth of annotated syllables. We then added more recordings for species with more data to investigate the influence of sample size. For *Melozone fusca*, we deliberately annotated (and trained on) a disproportionately small number of songs to generate artificial scarcity of data.

Annotations were performed in Raven Pro version 1.6.1 using a spectral window size (i.e., Fast Fourier Transform size) of 512 samples, which for a sample rate of 48,000 Hz was ~9.97 milliseconds (Fig 2). We chose 512 as it best balanced clarity between time and frequency in our spectrograms; it is also the default value in Raven Pro. We drew boxes representing frequency and time boundaries for every syllable in the recording of the focal species. These annotations were exported as text files using Raven's built-in selection format. After annotating the songs, we then subsetted the recordings such that individual recordings had approximately equal amounts of annotated song and silence. Preliminary results showed that when TweetyNet was trained on full files, which had a high proportion of non-annotated "silence", TweetyNet would fail to identify the songs and would only predict silence, but with high accuracy. As such, we wrote a script using tuneR in R which automatically sliced WAV files and their corresponding annotation files for training. Annotations that were separated by at least 1.0 seconds of non-annotated sound were split. Splits were made such that at ~50% of the resulting file was annotated: in cases where this was not possible, a larger percentage of the file may have been annotated. We then converted the annotation files into XML files that TweetyNet could use in the "birdsong-recognition-dataset" format following [81]. All annotations were given the label "1".

Annotated songs varied in the number of syllables per recording. Syllables per second rate ranged from 0.32 to 9.13 with an overall mean±standard deviation of 1.94±1.09. The mean ±standard deviation number of syllables per second ranged across taxa from 0.68±0.44 (*Myiarchus tuberculifer*) to 3.02±0.58 (*Passerina amoena*). Syllable lengths ranged from 0.02 to 1.99 seconds with a mean±standard deviation of 0.24±0.16. Syllables ranged in mean length from 0.14±0.05 (*Passerina amoena*) to 0.54±0.29 seconds (*Myiarchus tuberculifer*).

## Training of machine learning model

We benchmarked models according to established best practices for training and validating deep neural networks [40]. Training of TweetyNet models was done with the vak library version 0.6.0 [82]. We used version 0.8.0 of TweetyNet for this purpose. When training TweetyNet, we set our spectrogram parameters as follows: Fast Fourier Transform size of 512 to match our annotations in Raven, step size of 32 samples (~0.67 milliseconds), frequency cutoffs of 500 and 15,000 Hz which were the approximate boundaries of frequencies in our species, a log transformed spectrogram to correct for wide ranges of amplitude, and a minimum power log threshold of 6.25 to ignore background noise.

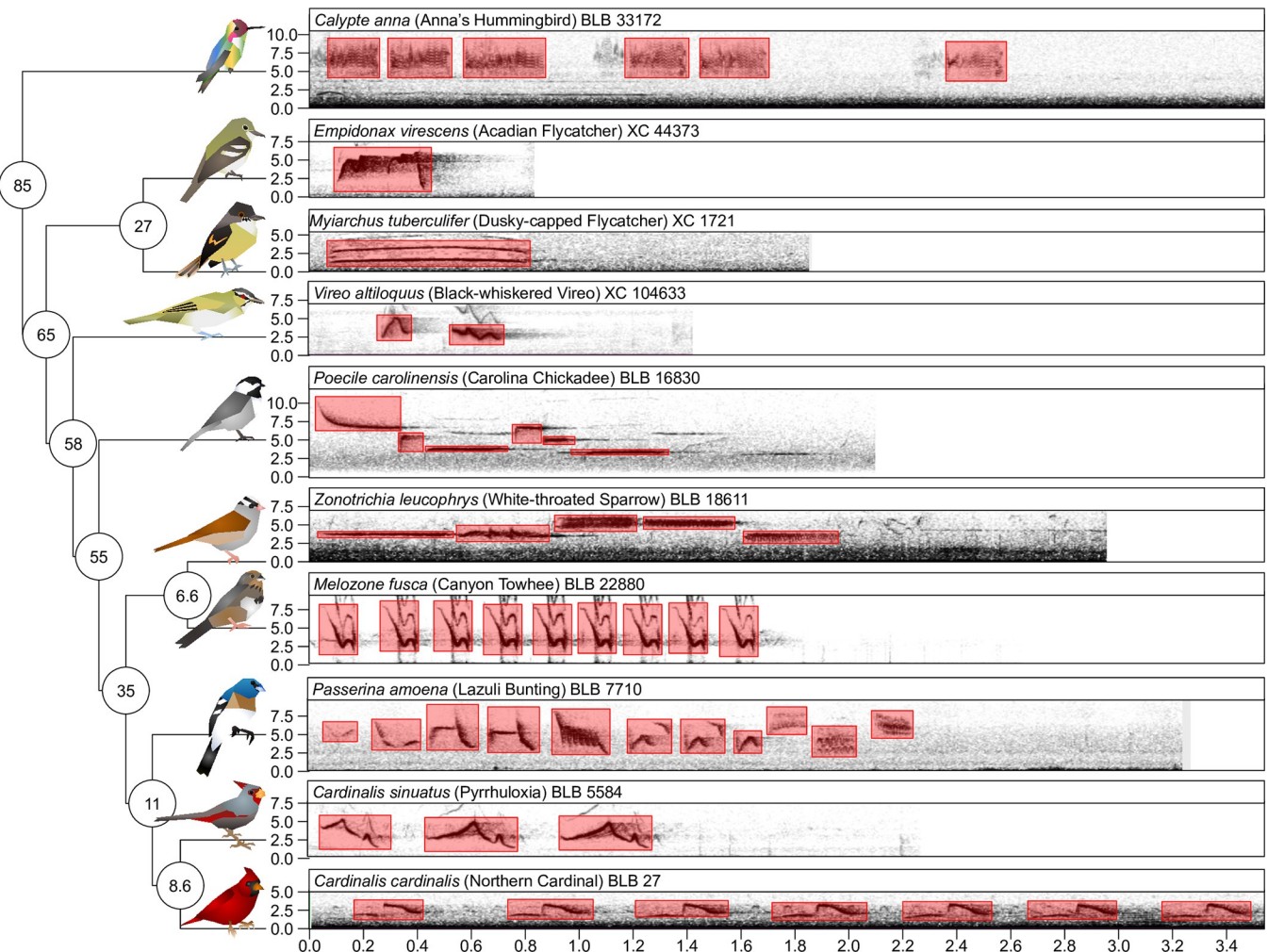

**Fig 2. Species in this study varied in their song types and phylogenetic relatedness.** On the left is a phylogeny describing the phylogenetic distances between ten focal species. Species are shown as artist's renditions of birds at tips of branches. Note that birds are not to scale. Numbers in circles at nodes show approximate time of divergence (in Mya; branches and nodes not to scale). On the right, spectrograms of exemplar songs per species are shown, with frequency on the y-axis (in kHz) and time on the x-axis (in seconds). Scientific, common names, and recording identification are shown above the spectrogram, where BLB = Borror Laboratory of Bioacoustics and XC = Xeno-Canto. Red boxes show the annotations that were manually put on per taxon.

We also used a window size of 88 spectral windows (the default for TweetyNet) and normalized spectrograms while training to account for differences in overall amplitude. Unless otherwise noted, we used an 80: 10: 10 split for training, validation, and test data, and tracked the total training time for each model. These splits were generated manually for each species. Each model used a batch size of 10; higher batch sizes caused runs to stop prematurely due to memory issues on the laptop we used. We validated the model every 400 steps (following [55] and consulting TweetyNet developers). Checkpoints were taken every 200 steps. We ran for 10 epochs (i.e., 10 passes the machine learning algorithm made over the full training dataset) but stopped training prematurely if the model had gone 100 checkpoint steps without an improvement in accuracy. We then used the maximum accuracy checkpoint as our final trained model. Some models were restarted if they were killed before completing 100 checkpoints without improvement. Models that failed this process multiple times were restarted from previous checkpoints and ran as needed.

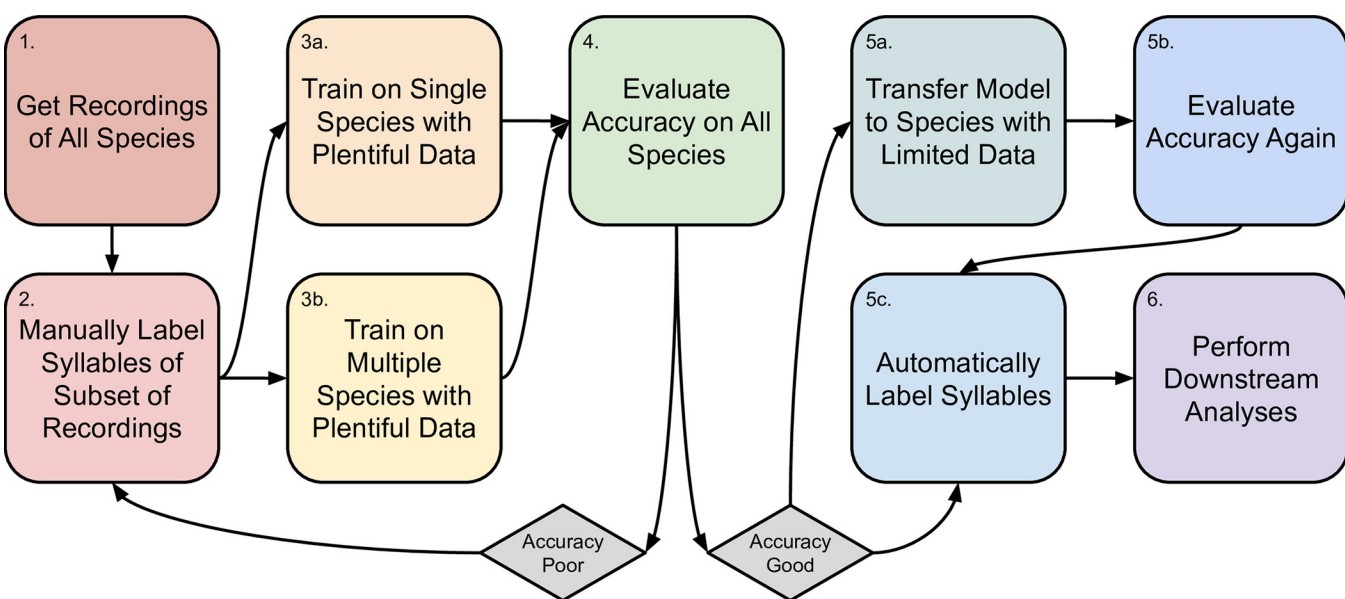

**Fig 3. Flowchart describing the procedure from acquiring song recordings to having segmented syllables for downstream analysis.** Boxes show general steps, arrows show flow of steps, and diamonds show decision point for data quality.

We trained three sets of models on the full datasets: single-species, multi-species, and fine-tuning (Fig 3). We did this once for each of our nine species with sufficient data, resulting in nine total single-species models. For the multi-species model, we trained a model in Tweety-Net as above but using a combination of the nine species with sufficient available data. We balanced the data in the multi-species model by species, taking subsets of the training and validation data such that each species had approximately the same amount of training and validation data (in seconds) as our species with the smallest sample size, to mitigate any effects of biased sampling. Preliminary results showed that not balancing the sample sizes across taxa resulted in a final model that was disproportionately affected by *Zonotrichia leucophrys*, which had the largest dataset. We ran this model until it had converged for 400 time steps instead of 100 like the other models because our preliminary results also showed that this model was more unstable as it trained.

After training the multi-species model as above, we then used that finished checkpoint to train on data from *Melozone fusca*, which we manipulated to use a relatively small amount of data. Though we had a comparable amount of data for this species compared to the other nine, we generated artificial scarcity by only using ~30% of this as training and validation data, with the remainder acting as test data [83], to understand how this impacted performance. This resulted in one single fine-tuned model.

We tested all models on annotated data from each of the ten species (including *Melozone fusca*) as well as the combination of all species used in the multi-species model. Before evaluating the performance of models, we removed any predicted syllables that were less than 0.02 seconds in length, as this was the length of our shortest syllable in our dataset (following [55]). In all cases, we assessed the accuracy, precision, recall, and F-score of the model. Whether a given ~0.67 millisecond window was predicted correctly as a syllable or not determined these four performance metrics. Accuracy is the percentage of positively identified syllables. Precision is the number of true positives over true and false positives. Recall is the number of true positives over the number of true positives and false negatives. F-score is a weighted average of precision and recall.

We also assessed the segment error rate (also known as syllable error rate), which is the number of predicted syllables minus the number of observed syllables divided by the number of observed syllables. From this we calculated absolute segment error rate, agnostic to whether there were more or fewer predicted than observed syllables; we use this latter performance metric unless otherwise specified. All performance metrics were calculated for each recording and then summarized as an overall value per species, plus a mean and a deviation across subsets of recordings. TweetyNet also outputs accuracy and segment error rate metrics; however, this is done before the removal of short predicted syllables. These values are highly correlated with, but not the same as, their corresponding manual calculations (see S7 Table in S1 File).

We evaluated performance across species with respect to the predicted genetic distance. In addition, we evaluated the performance with respect to the number of species present in the training dataset and whether fine-tuning was used. For some analyses we examined the phylogenetic distance between trained species and test species. We mined these data from timetree.org [84], taking the estimated time listed as the time of divergence in millions of years (Mya; Fig 2). Intraspecific comparisons were given an estimated time of 0.0 Mya; we also did this for the fine-tuning done with *Melozone fusca*. For the multiple species model, we calculated the weighted average estimated divergence time based on the sample sizes for each species, again taking conspecifics as 0.0 Mya.

To investigate how our sample sizes impacted our results, we evaluated the performance on our data using different sample sizes with the 'learncurve' functionality in vak. This artificially subsets the training data into different pre-specified sizes to assess accuracy with the reduced dataset. We subset the data such that models were trained with 10, 50, 100, 150, 200, 250, 500, 1000, and 2000 seconds of song. Note that because sample sizes varied across species, not all taxa were trained with the higher subsets. For each sample size, we generated 10 datasets per species and trained a model on them with the above parameters. Because these analyses involved running the same models hundreds of times, we ran them on the Ohio Supercomputing Center cluster Pitzer [85]. Anecdotally, we found that models of the same size ran much faster and with fewer errors in a high-performance computing setting.

## Song diversity

We evaluated song diversity to determine whether this impacted our machine learning model accuracy, as we suspected that species with more diverse songs would be more difficult to segment. Song diversity was evaluated in two ways: first, we consulted an analysis that focused on Passeriformes which measured song variation and complexity [76]. Seven of the species were analyzed in that study (excluding *Calypte anna*, *Melozone fusca*, and *Empidonax virescens*) which were calculated by performing a principal components analysis (PCA) on song properties. We directly used those values as measures of within-individual variation, within-species variation, and complexity (i.e., repetition within songs).

Our second metric of song diversity was calculated by estimating the hypervolume of syllables used for each species. To do this, we took our manually annotated recordings and put them through the SoundShape pipeline [86]. SoundShape aligns and normalizes syllables of sound to make them directly comparable with respect to their size and shape. After aligned and normalized syllables were converted to TPS format, we randomly selected 50 syllables per species and calculated a PCA on their values, retaining sufficient principal components (PCs) to explain at least 50% of the variation (range: 16–20 PCs of 500, mean = 17.66). We used the function 'prcomp' in the stats version 4.2.1 package in R version 4.2.1 [87]; note that this later version reflects time between downloading and processing recordings. For each retained PC, we calculated a mean and standard deviation for each species and then calculated a pseudo-hypervolume as the product of the standard deviations. Preliminary analyses showed that formally calculated

hypervolumes (with the 'convhulln' function in geometry version 0.4.6.1 package in R) are prohibitively expensive with respect to analysis time when calculating a ~20-dimensional hypervolume. However, our pseudo-hypervolumes are highly correlated with the formal hypervolumes up to 7 PCs (see S4 Table in S1 File), so we use the pseudo-hypervolume as a proxy (hereafter hypervolume). We repeated our PCA and hypervolume calculations 50 times, standardizing hypervolumes such that the species with the largest hypervolume had a relative hypervolume of 1.0, and then from the 50 repetitions we extracted the mean and standard deviation of the relative hypervolume per species to use as our diversity metric. Species with more diverse syllable types, and therefore more song diversity, should have larger hypervolumes.

## Impact of various metrics on performance and training time

To investigate what aspects of our models determined performance and training time, we used linear regressions [88] using the 'lm' function in the stats package in R. We took the log values of training time and sample size, as these varied by orders of magnitude across taxa and models. We used a Bonferroni correction to set our alpha level to 0.000413 (0.05/121 for 11x11 model comparisons) to determine significance [89]. All linear models were univariate, with only a single predictor and response variable per model. For a complete list of models and their parameters and coefficients, see the S1-S7 Tables in S1 File.

We tested the relationships between the estimated divergence time between trained and tested taxa, song diversity metrics, and performance metrics with linear models. We calculated correlations between our song diversity metrics with linear models. To test performance vs song complexity, we took the mean performance value across species, taking means across training datasets as well as testing datasets.

Finally, we used ANOVA models [90] to determine whether individual species differed in performance metrics using the 'aov' function in the stats package in R. We evaluated performance given species identity to determine whether there were differences overall, and then interrogated which species were significantly different from each other. For this latter analysis we used Tukey's Honest Significant Differences tests [91] via the 'TukeyHSD' function in the stats package in R.

## Song data summary

Our dataset included 6,563 *Zonotrichia leucophrys* songs, 918 *Cardinalis cardinalis* songs, 36 *Cardinalis sinuatus* songs, 128 *Empidonax virescens* songs, 103 *Calypte anna* songs, 72 *Melozone fusca* songs, 148 *Myiarchus tuberculifer* songs, 427 *Poecile carolinensis* songs, 12 *Vireo altiloquus* songs, and 62 *Passerina amoena* songs. We then annotated 201 *Zonotrichia leucophrys* songs, 63 *Cardinalis cardinalis* songs, 8 *Cardinalis sinuatus* songs, 22 *Empidonax virescens* songs, 23 *Calypte anna* songs, 12 *Melozone fusca* songs, 34 *Myiarchus tuberculifer* songs, 32 *Poecile carolinensis* songs, 12 *Vireo altiloquus* songs, and 44 *Passerina amoena* songs.

After balancing the relative amount of annotated sound to silence, this resulted in total datasets of 4,380 seconds for *Zonotrichia leucophrys*, 1,091 seconds for *Cardinalis cardinalis*, 400 seconds for *Cardinalis sinuatus*, 291 seconds for *Empidonax virescens*, 886 seconds for *Calypte anna*, 272 seconds for *Melozone fusca*, 314 for *Myiarchus tuberculifer*, 395 for *Poecile carolinensis*, 321 for *Vireo altiloquus*, and 320 for *Passerina amoena* (see S1 Table in S1 File).

## Results

### Machine learning model performance on annotated song

First, we determined how training time of the models in a laptop setting depended on the amount of data. Training time for full single-species models ranged from 19,025 seconds

**Table 1. Models trained in TweetyNet across multiple species vary in training time, dataset size, and type of model.** "9 Species" models were trained on all data from the "Single species" type models.

| Species Trained On | Training Time (sec) | Training: Val: Test (sec) | Type |
|---|---|---|---|
| *Calypte anna* | 57,232 | 554: 69: 70 | Single species |
| *Cardinalis cardinalis* | 52,407 | 867: 108: 108 | Single species |
| *Cardinalis sinuatus* | 23,782 | 320: 40: 40 | Single species |
| *Empidonax virescens* | 41,314 | 231: 29: 29 | Single species |
| *Myiarchus tuberculifer* | 24,185 | 251: 31: 31 | Single species |
| *Passerina amoena* | 19,025 | 256: 31: 31 | Single species |
| *Poecile carolinensis* | 29,818 | 313: 40: 40 | Single species |
| *Vireo altiloquus* | 22,807 | 257: 31: 32 | Single species |
| *Zonotrichia leucophrys* | 48,532 | 3768: 465: 482 | Single species |
| 9 Species | 125,134 | 2078: 270: 1175 | Multiple species |
| 9 Species+*Melozone fusca* | 37,429 | 71: 9: 192 | Fine tuning |

(~5 hours 17 minutes, *Passerina amoena*) to 57,232 seconds (~15 hours 53 minutes, *Calypte anna*). Notably, despite having an order of magnitude more data to train over than any other single species model, training time for the largest dataset (*Zonotrichia leucophrys*) was lower than expected at 48,532 seconds (~13 hours 28 minutes). The multiple-species model took 125,134 seconds to run (~34 hours 45 minutes) but reached the 100-step checkpoint at 45,461 seconds (~12 hours 37 minutes), and the fine-tuning model built from it took 37,429 seconds to run on the *Melozone fusca* dataset (~10 hours 23 minutes). The relationship between the size of the training dataset and the amount of training time needed is positive on a log scale (adjusted $R^2 = 0.37$, $p = 5.4 \times 10^{-14}$; Table 1; S7 Table in S1 File; Fig 4). Anecdotally, models that needed to be restarted multiple times (due to memory issues, etc.) took much longer to converge, likely because restarting the model also reset the number of iterations since improvement (aka patience) to 0 of 100.

Performance of our models across accuracy, precision, recall, and F-score varied across species with respect to phylogenetic distance between training and testing datasets. For models that were trained on single species, accuracy was generally highest when models were tested on the same species that they were trained on (range = 0.92–0.97), except for *Cardinalis cardinalis* in which accuracy was highest for its congener *Cardinalis sinuatus* (0.93 vs 0.92) and *Passerina amoena* which performed equally well on *Empidonax virescens* (0.92 for both). Accuracy dropped off in a manner that scaled to the predicted phylogenetic distance in the models that were not trained and tested on the same taxa. The congeneric comparisons have one comparison with a 1% gain and one with a 7% loss in accuracy. Taxa with the same oscine/suboscine status show a loss of 0–21%, in the same order show a loss of 0–29%, and comparisons between passerines and non-passerines show a loss of 12–43%. This relationship is significantly negative with respect to estimated phylogenetic distance (adjusted $R^2 = 0.30$, $p < 3.5 \times 10^{-11}$; Fig 5; S7 Table in S1 File). Precision, recall, and F-score behaved almost identically to accuracy, where each was highly negatively correlated with phylogenetic distance (adjusted $R^2 > 0.22$, $p < 2.51 \times 10^{-8}$; S7 Table in S1 File). Precision ranged from 0.33–0.99, recall ranged from 0.03–0.95, and F-score ranged from 0.06–0.96.

Performance via segment error rates show similar patterns to accuracy measures. Before taking absolute values, average segment error rates were usually positive (only 10/121 were negative) indicating that more segments were produced than originally predicted. After taking absolute values, error rates ranged from 11.9%–592.6%. Error rates for models trained and tested on the same species ranged from 11.9% (*Passerina amoena*) to 145.8% (*Calypte anna*).

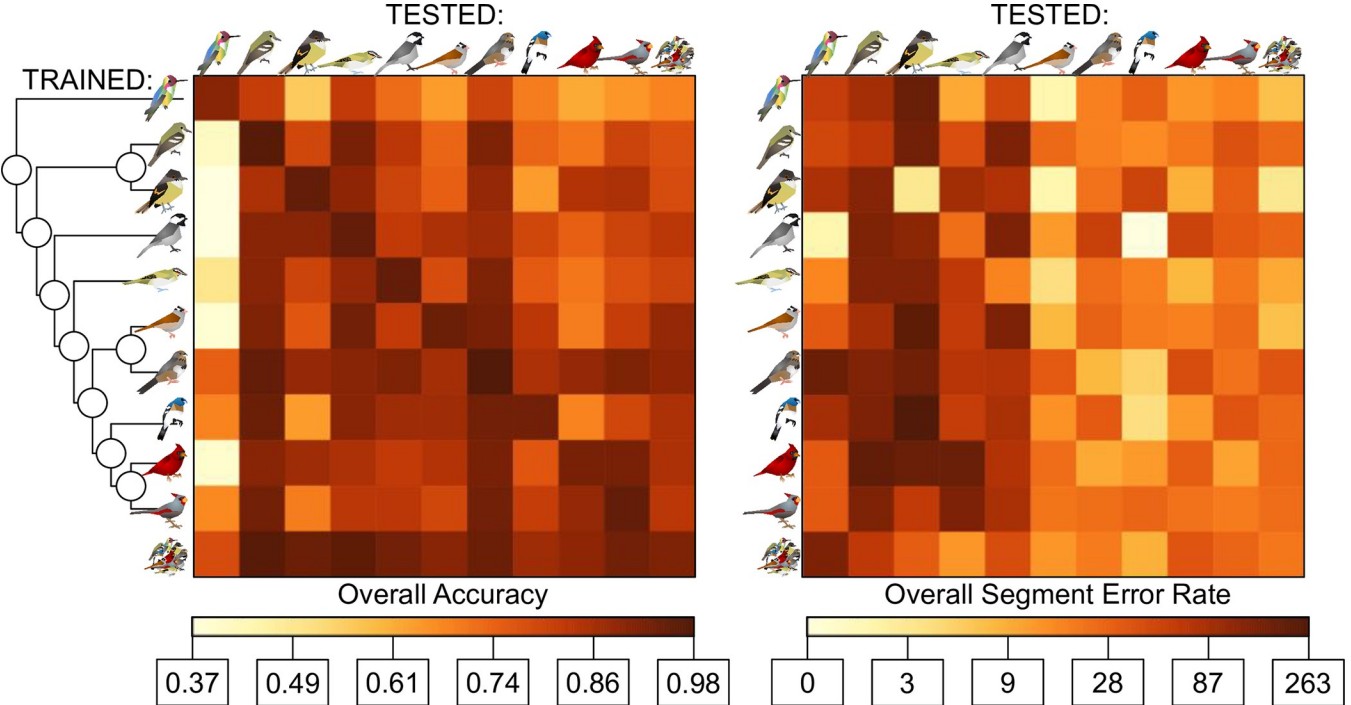

**Fig 4. Frame rate accuracy and syllable error rate of models trained in TweetyNet across species.** Far left shows phylogeny from Fig 1 with species at tips. Left: frame rate accuracy between species trained on (top to bottom) and species tested on (left to right). Species given with same symbols as in Fig 1 plus all nine species (circular cluster). For accuracy, darker values indicate higher accuracy. Right: segment error rate between species trained on and tested on. For segment error rate, darker values indicate higher error (worse performance) on a log scale, where 0 = 0% error and 263 = 263% error.

Models tested on *Myiarchus tuberculifer* and *Calypte anna* have significantly higher error rates than all other species (p = 5x10$^{-15}$, df = 110, f = 13.3, sum squares = 551384, mean squares = 55138). Segment error rates were usually lowest when models were trained and tested on the same taxa; exceptions include *Calypte anna* (where the *Myiarchus tuberculifer*

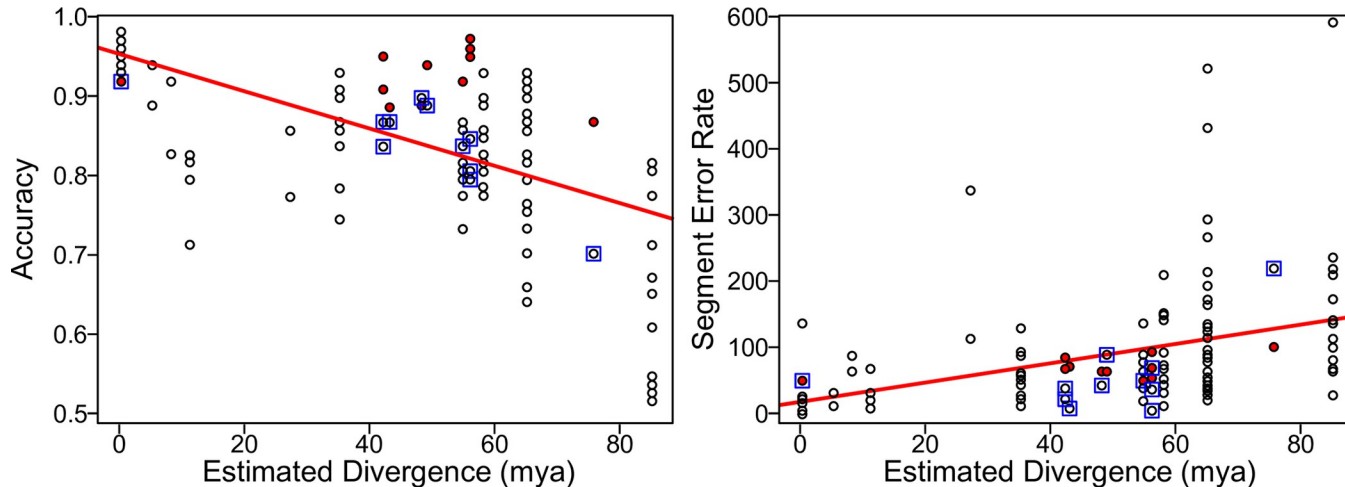

**Fig 5. Model performance by species drops with estimated divergence, but training on multiple species performs better than expected given average divergence time.** X-axis gives the estimated divergence time. Y-axis gives the accuracy (left) or segment error rate (right). Hollow black points show single-species models. Filled red points show multiple-species models. Blue square outlines show models where the test dataset was all species simultaneously. Red line shows the overall line of best fit (see S1 and S2 Figs in S1 File for individual species).

model performed best, 92.4% vs 145.8%), *Cardinalis sinuatus* (where the congeneric *Cardinalis cardinalis* model performed best, 22.6% vs 26.7%); and *Vireo altiloquus* (where the full multi-model performed best, 17.9% vs 21.8%). Segment error rate is not significantly associated with phylogenetic distance (adjusted $R^2 = 0.07$, $p = 0.0017$; S7 Table in S1 File), although the relationship is still positive such that error rates are higher with more phylogenetic distance.

Performance metrics were generally positively correlated. Model accuracy, precision, recall, and F-score were all highly correlated (adjusted $R^2 > 0.14$, $p < 9.95 \times 10^{-6}$; S7 Table in S1 File) except for precision and recall with each other which were not significantly correlated (adjusted $R^2 = 0.04$, $p = 0.012$). Segment error rates were significantly negatively correlated with precision (adjusted $R^2 = 0.22$, $p = 5.07 \times 10^{-8}$). This negative correlation is expected, as higher segment error indicates lower performance, whereas higher accuracy, precision, recall, and F-score indicate higher performance. However, segment error was not correlated with the other variables (adjusted $R^2 < 0.02$, $p > 0.06$).

Irrespective of phylogenetic distance, we also found that *Calypte anna* vocalizations were significantly less likely to be segmented accurately than other species. Notably, this is the only non-passerine in the dataset and has a very different type of song than the others. For accuracy (sum squares = 1.001, $p = 1.57 \times 10^{-12}$), and recall (sum squares = 2.018, $p = 1.04 \times 10^{-5}$; S2 Table in S1 File), *Calypte anna* has lower values than some if not most other taxa. This relationship shows the same trend but is not significant for F-score (sum squares = 1.106, $p = 0.00061$). For precision, this trend is the same pattern not significant for *Calypte anna*, but *Empidonax virescens*, a non-learning suboscine, does show lower precision than other species (sum squares = 0.9731, $p = 1.03 \times 10^{-6}$).

Although phylogenetic distance is a significant predictor, there is no significant association between sample size of either training, validation, or test data and average accuracy across models (log sample sizes: adjusted $R^2 < 0.073$, $p > 0.001$; S7 Table in S1 File). This corroborates evidence from the learncurve analyses: in the latter, there was a positive relationship between the size of the dataset used for training and accuracy, although there were diminishing returns to the gain in accuracy as the training set size increased (S3 and S4 Figs in S1 File). Models plateaued quickly; accuracy on the full datasets had a mean±standard deviation of 0.94±0.02, whereas accuracy on 50 seconds was 0.90±0.04 and accuracy on 200 seconds was 0.93±0.02. Overall, this suggests that our datasets were sufficient to get accurate segmentation estimates.

Models that were trained on multiple species, rather than single species, behaved similarly. The multi-species model performs comparably to models trained and tested on the same species, ranging from a 5% loss in accuracy to a 5% gain in accuracy (mean of 1% gain in accuracy). However, for any given species the multi-species model always outperforms models trained and tested on different species (with the next best model having a 2%–10% loss in accuracy). They also perform better than expected for their mean estimated divergence time, consistently being above the line of best fit with respect to accuracy and on or lower than the line of best fit when it comes to segment error rate (Fig 5).

## Impact of small amounts of data on training and fine-tuning

Next, we wanted to understand how our approach performed in a scenario when researchers only have a small amount of data, using our artificially-data-deficient *Melozone fusca* dataset. With respect to the small *Melozone fusca* dataset, segmentation of this species' songs is overall highly accurate (Table 2, Fig 6). For the single-species models, the two least accurate species were the distantly related *Calypte anna* and the suboscine *Empidonax virescens* (0.77 for both). The most accurate single-species models were *Cardinalis sinuatus* and *Passerina amoena* (0.92 for both). The multi-species model was highly accurate (0.93), suggesting that having a model

**Table 2. Performance of models trained in TweetyNet for *Melozone fusca*.**

| Train | Test | Acc | Prec | Rec | F | S.E.R. | Div |
|---|---|---|---|---|---|---|---|
| 9 Species | M. fusca | 0.94 (0.94+0.05) | 0.87 (0.79+0.26) | 0.91 (0.89+0.09) | 0.89 (0.81+0.20) | 0.19 (1.00+1.46) | 49 |
| C. anna | M. fusca | 0.80 (0.82+0.15) | 0.65 (0.69+0.32) | 0.37 (0.46+0.31) | 0.47 (0.49+0.28) | 0.18 (1.50+2.93) | 85 |
| C. cardinalis | M. fusca | 0.92 (0.91+0.05) | 0.77 (0.69+0.27) | 0.95 (0.93+0.08) | 0.85 (0.76+0.21) | 0.10 (1.38+1.85) | 35 |
| C. sinuatus | M. fusca | 0.93 (0.93+0.05) | 0.83 (0.76+0.27) | 0.89 (0.89+0.08) | 0.86 (0.79+0.20) | 0.23 (1.03+1.95) | 35 |
| E. virescens | M. fusca | 0.91 (0.91+0.06) | 0.86 (0.80+0.26) | 0.77 (0.79+0.19) | 0.81 (0.75+0.19) | 0.18 (0.96+1.17) | 65 |
| 9 Species +M. fusca | M. fusca | 0.98 (0.98+0.02) | 0.97 (0.96+0.09) | 0.95 (0.93+0.08) | 0.96 (0.94+0.07) | 0.08 (0.18+0.53) | 0 |
| M. tuberculifer | M. fusca | 0.88 (0.88+0.06) | 0.82 (0.73+0.32) | 0.66 (0.65+0.20) | 0.73 (0.66+0.22) | 0.22 (1.05+1.52) | 65 |
| P. amoena | M. fusca | 0.93 (0.93+0.05) | 0.84 (0.78+0.29) | 0.89 (0.90+0.11) | 0.87 (0.79+0.22) | 0.32 (0.99+1.43) | 35 |
| P. carolinensis | M. fusca | 0.91 (0.92+0.06) | 0.96 (0.88+0.23) | 0.68 (0.62+0.23) | 0.80 (0.71+0.20) | 0.24 (0.59+0.74) | 55 |
| V. altiloquus | M. fusca | 0.87 (0.85+0.15) | 0.69 (0.63+0.38) | 0.86 (0.78+0.29) | 0.77 (0.70+0.29) | 0.55 (2.19+3.32) | 58 |
| Z. leucophrys | M. fusca | 0.92 (0.92+0.05) | 0.86 (0.77+0.32) | 0.82 (0.68+0.24) | 0.84 (0.69+0.28) | 0.28 (0.98+1.65) | 8 |

Numerical estimates for performance (Acc, Prec, Rec, F, S.E.R) are given as point estimate (mean+standard deviation). "Train" = species model was trained on. "Test" = species model was tested on. "Acc" = model accuracy. "Prec" = model precision. "Rec" = model recall. "F" = model F-score. "S.E.R" = model segment error rate. "Div" = divergence time between trained and tested species. Estimated divergence time for "9 Species" model is a weighted average, proportional to the sample sizes used for each species trained on.

trained on a more diverse subset of sounds improves performance overall. Lastly, the fine-tuned model performed the best of them all with comparable additional training time (0.96), suggesting that having a single multi-purpose model rich in data to "seed" the training of species with few data may be beneficial.

*Calypte anna* had the lowest precision (0.65), recall (0.37), and F-score (0.47) of the single-species models on *Melozone fusca*. Which single-species model had best performance varied, with *Poecile carolinensis* having the best precision (0.96), *Cardinalis cardinalis* having the best

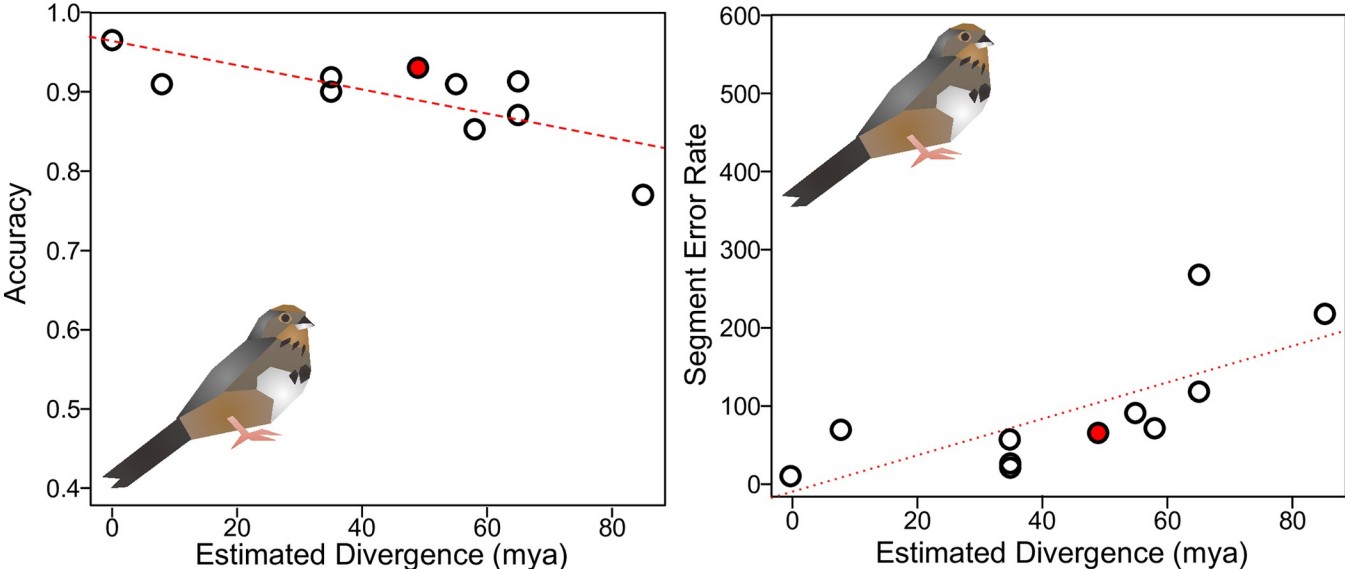

**Fig 6. Model performance on *Melozone fusca* drops with estimated divergence, but also shows recovery when trained on multiple species.** X-axis gives the estimated divergence time. Y-axis gives the accuracy (left) or segment error rate (right). Hollow black points show single-species models. Filled red points show multiple-species models. Red line shows the overall line of best fit. Solid line shows a significant relationship, while dotted line is not significant (see S8 Table in S1 File).

**Table 3. Species vary with respect to song complexity, song variation, and hypervolume diversity.**

| Species | Complexity[1] | Individual Variation[1] | Species Variation[1] | Hypervolume Mean±SD |
|---|---|---|---|---|
| *Calypte anna* | n/a | n/a | n/a | 0.8607±0.2325 |
| *Poecile carolinensis* | 0.665 | -1.089 | -0.970 | 0.4105±0.4101 |
| *Zonotrichia leucophrys* | -0.695 | -0.494 | 0.567 | 0.1468±0.2273 |
| *Myiarchus tuberculifer* | 1.541 | -0.133 | -0.228 | 0.0538±0.1473 |
| *Passerina amoena* | -0.106 | -0.248 | -0.550 | 0.0304±0.0403 |
| *Vireo altiloquus* | 1.586 | 0.025 | -1.098 | 0.0229±0.0387 |
| *Empidonax virescens* | n/a | n/a | n/a | 0.0182±0.0425 |
| *Melozone fusca* | n/a | n/a | n/a | 0.0019±0.0028 |
| *Cardinalis cardinalis* | -1.798 | 0.421 | 1.872 | 0.0009±0.0004 |
| *Cardinalis sinuatus* | -1.086 | 1.416 | 0.955 | 0.0002±0.0003 |

Within-individual and within-species variation scores range from lower variation (more negative) to higher variation (more positive). Complexity scores range from lower complexity/more repetition (more negative) to higher complexity/less repetition (more positive). Hypervolume diversity is a percentage of total hypervolume across all six species, with means and standard deviations from 50 independent runs of 50 random syllables per species.

[1]from [76]

recall (0.95), and *Passerina amoena* having the best F-score (0.87). The multi-species model generally had medium-to-high values for performance (0.87 precision, 0.91 recall, 0.89 F-score) while the fine-tuned model always performed the best of any model (0.97 precision, 0.95 recall, 0.96 F-score). Segment error rates varied across models as well. Of single-species models, *Cardinalis cardinalis* had the lowest error rate (10%), while *Vireo altiloquus* had the highest (55%). The multi-species model had medium error rates (19%) and the fine-tuned model once again outperformed all others (8%).

## Song diversity results

Given our results above, we tested if there was a relationship between performance and song diversity, as we expect that higher diversity songs should be harder to segment. Song complexity values (from [76]) ranged such that *Cardinalis cardinalis* had the most negative complexity and *Vireo altiloquus* had the most positive complexity (Table 3). Individual variation (from [76]) ranged from *Poecile carolinensis* with the most negative to *Cardinalis sinuatus* with the most positive. Species variation (from [76]) ranged from *Vireo altiloquus* with the most negative to *Cardinalis cardinalis* with the most positive. Our calculated hypervolume diversity metric finds that instead *Cardinalis sinuatus* has the smallest hypervolume with *Zonotrichia leucophrys*, *Poecile carolinensis*, and *Calypte anna* having the largest hypervolumes. The latter three species occupy much higher percentages of hypervolumes compared to the remaining four species. Mean hypervolume is positively correlated with all three of the complexity values from [76], but it is especially correlated with individual variation (Fig 7; S5 and S6 Tables in S1 File).

Models are less precise when tested on species that are more complex (adjusted $R^2 = 0.81$, $p = 0.0032$) and are less accurate (adjusted $R^2 = 0.52$, $p = 0.012$) and have worse F-scores (adjusted $R^2 = 0.44$, $p = 0.020$) when tested on species that have larger hypervolumes (S5 and S6 Tables in S1 File). This suggests that species with more complicated sounds, either because of a larger diversity of syllables overall or more propensity to vary their syllables during single songs, are harder to segment for any given model.

Irrespective of which species they are tested on, models are less accurate when trained on species with higher complexity (adjusted $R^2 = 0.66$, $p = 0.015$) or larger hypervolumes (adjusted $R^2 = 0.60$, $p = 0.0047$), less precise when trained on species with more individual

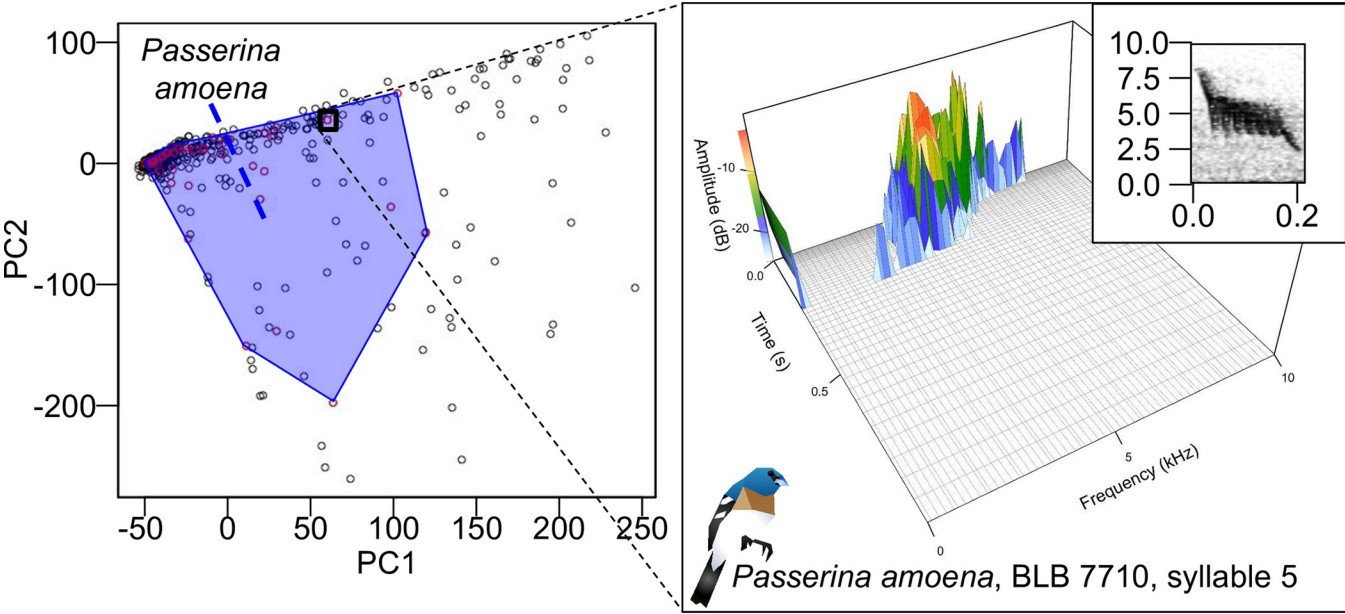

**Fig 7. Example hypervolume diagram for one species, *Passerina amoena*.** Left: principal component analysis of SoundShape for 500 syllables (50 per 10 species); with PC1 (x-axis) and PC2 (y-axis). *Passerina amoena* syllables are shown with red points and a 2-dimensional convex polygon is given to display the 2-dimensional hypervolume. Hypervolume calculation was done 50 times with 50 syllables per species. Right: SoundShape diagram for point indicated in black box on left plot, representing syllable 5 from BLB 7710. X-axis shows the frequency of syllables (kHz). Y-axis shows time (seconds); note this is on a log scale. Z-axis shows amplitude; amplitude is also color-coded on the plot with warmer colors indicating louder amplitude. Inset: original spectrogram representation (see Fig 2).

variation (adjusted $R^2$ = 0.52, p = 0.039), and have worse F-scores when trained on species with larger hypervolumes (adjusted $R^2$ = 0.60, p = 0.0047; S5 and S6 Tables in S1 File). Recall follows these trends such that species with larger hypervolumes (adjusted $R^2$ = 0.47, p = 0.016) and more variable hypervolumes (adjusted $R^2$ = 0.35, p = 0.040) have worse recall, but models trained on species with more individual variation had better recall (adjusted $R^2$ = 0.65, p = 0.016). This suggests that overall, models that are trained on species with more syllable diversity and complexity are more likely to identify notes that are not present; anecdotally, many of the syllables that are identified in this way tend to be from another bird species present in the recording (e.g., extra syllables in Fig 8).

## Discussion

Numerous initiatives have been undertaken in the past decades to accumulate and archive biological information into searchable and accessible databases. Such efforts include databases of morphological characteristics (MorphoNet; [92]), localities for use in modeling abundance and niche (GBIF; gbif.org), genetic, genomic, and protein sequences (GenBank; [93]), and of course bioacoustics. Taking full advantage of all these datasets requires increases in efficiency, including development of automated retrieval of data as well as data extraction.

Bioacoustic data have traditionally been parsed manually since spectrograms were developed in the 1950s [94]. These methods have required large amounts of time, both to extract the usable data but also to verify and validate the homologous aspects of songs for comparisons within and between taxa. Songs must be searched for the actual sounds, and then researchers must decide which parts of the sound are considered unique syllables, before even beginning to classify them by which type of syllable is being produced. Often this necessitates deep understanding of the species in question, and while this approach is inherently valuable and has

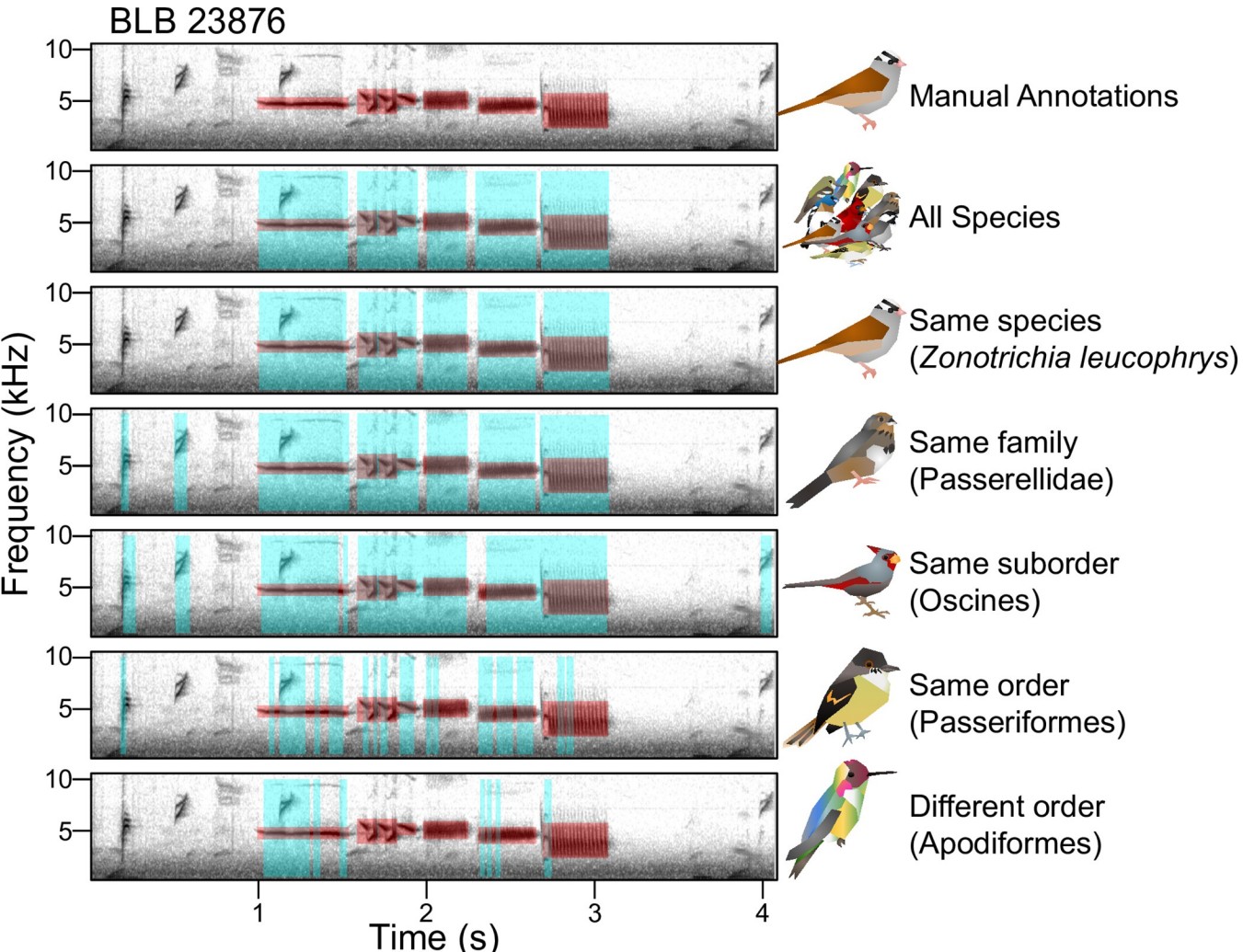

**Fig 8. Performance of models on the same data vary between training species.** Spectrogram depicts recording of *Zonotrichia leucophrys* BLB 23876, with time (seconds) on the x-axis and frequency (kHz) on the y-axis. Top row: the seven manual annotations used as the "true" segmentation in red; note the third through fifth syllable are separated by very small gaps and that there are notes from other species unlabeled as background noise. The following rows from top to bottom are predictions output from models trained on all species (9 Species Balanced), the same species (*Zonotrichia leucophrys*), another species from the same family (*Melozone fusca*), another species from the same suborder (*Cardinalis sinuatus*), another species from the same order (*Myiarchus tuberculifer*), and finally a species from a completely different order (*Calypte anna*). Predictions are given in blue with "true" values superimposed in red.

provided the foundation for avian bioacoustics, it may preclude a broader understanding of vocalizations outside of those focal taxa. Automated methods promise to improve the efficiency of bioacoustics by identifying data more readily and by automatically calculating differences between sounds [e.g., 86]. Despite these improvements, the attributes and limitations of these methods are not as well understood in comparison to more traditional manual analyses. One aspect that has been investigated is the amount of time spent: processing songs in automated ways can be up to ~5 times faster compared to manual methods [30].

## Evaluation of performance

Though the final sample size does not appear to impact performance, it is an important consideration when trying to decide how to best extract data from species. With sufficiently large

sample sizes (TweetyNet suggests three minutes of annotation, [55]), techniques like convolutional neural networks will work effectively to learn and parse the diversity of syllables and sounds present in these data. However, for some taxa the available data are not sufficient for these types of learning. In this case, we have two recommendations depending on the study system. If a study only concerns one single species without a large amount of data, it would likely be best to manually annotate all data available, rather than try to fit a computationally expensive model to it. On the other hand, if a study is focused on multiple species at once, especially if some of those species have a large sample size, we recommend training a model on multiple different species, optionally then using transfer learning techniques to fine-tune the model. Learning from multiple species substantially increases accuracy for taxa without a large amount of data, as models trained on multiple species perform better than models trained on single species when applied to a never-before-seen taxon. The caveat with this latter approach is that the benefit of this learning may decrease as phylogenetic distance (and song dissimilarity) increases with respect to your focal taxa. With respect to computational efficiency, training a model with multiple different species takes less time than training a model on each species separately, which our preliminary results suggest is because of difficulty converging. We suspect this is because the algorithm needs to optimize segmentation of many disparate kinds of syllables, much like with species that have more complex songs compared to species with fewer songs. Despite the increase in computation time, having a more general-purpose, multi-species model trained can be worthwhile for researchers looking to maximize the number of taxa they can segment syllables for, while minimizing the amount of manual annotation and training time they need.

Training a model on a species and applying it to that same species appears to be the most accurate method of extracting syllables, provided that the sample size is sufficiently high. This is irrespective of whether the training dataset includes only that species or also includes other taxa (e.g., the 9-species models). However, the range in accuracy varies highly across taxa. Species with relatively simple vocalizations (e.g., *Empidonax*) appear more likely to be segmented with high accuracy than species with more diverse vocalizations (e.g., *Zonotrichia*). We speculate that this is due to a combination of factors. Taxa with more complex vocalizations must simultaneously optimize the detection algorithm for multiple different syllable types, which is expected to be more difficult than categorizing fewer kinds of syllables. Our models were run under the same parameters but were allowed to exit early if they reached an accuracy plateau. Given this, it may be that the convolutional neural network can solve a "simpler" problem in the same amount of time more efficiently. Nearly every song diversity metric we used shows this relationship to some degree. However, with recall specifically, models trained on more diverse species are more likely to have high recall, returning a high proportion of the syllables correctly at the potential expense of many false positives. From examining the data, we suspect this is because these over-predict syllables, particularly by identifying background sounds from non-focal species as being valid syllables. Though this may not be optimal behavior for every use case, this nevertheless suggests to us that training on species with more variation may be better for downstream applications because it could allow the models to detect new, unseen syllables more readily than models that are highly precise. Another consideration is related to the sample size given to the model. It is possible that our high accuracy on these species is due to over-fitting, rather than any biological aspect of the song. However, we do not attribute these differences to learning vs. non-learning behavior; although *Empidonax* and *Myiarchus* are the only suboscine (non-learning) Passeriform birds in our dataset, there is not a relationship between song complexity and oscine/suboscine status in Passeriformes [74].

## Phylogenetic and ecological implications

Model performance drops with phylogenetic distance between the training data and tested data. This suggests that there is commonality between closely related species in terms of their song; indeed, evidence across passerines suggests that song is phylogenetically constrained, in addition to being influenced by morphology and ecology [74, 75]. From these results we would predict that any similar learning algorithm should behave in this fashion with traits that are associated with phylogeny. This pattern may also extend to species with convergent song features as well, though this remains to be tested.

The structure of bird songs—namely the complexity, variation, and diversity of syllables used—impacts segmentation, with more complex, variable, and diverse sounds being harder to segment accurately. The evolution of complexity (in a broad sense) has multiple proposed mechanisms, where songs can become more complex due to environments selecting for sound transmission [95, 96], phenological constraints on breeding behavior influenced by migration [97], but see [98], or sexual selection driving songs to become more challenging to produce and thus more effective signals of quality [99]. However, complexity is poorly defined across studies, though it generally encapsulates variability, diversity, and repertoire size, [11, 98]. The metrics of complexity we used via [76] encapsulate variability (individual- and species-level variation in frequency, song length, and syllable number) as well as repertoire size (complexity as proportion of unique notes, song types, and repetition of syllables). Our hypervolume metrics encapsulate variability (standard deviation hypervolume) and repertoire size (mean hypervolume) at the same time. We recommend that for researchers focusing on single taxa with very diverse vocalizations or multiple taxa at once, model performance be especially scrutinized, and additional data labeling or training time be used to increase overall accuracy.

Evaluating the role of phylogeny or ecology in structuring bird song is beyond the scope of this paper. However, given that phylogeny seems to have a direct impact on performance, and ecology may have an indirect impact (via the relationship between environment and complexity), we believe that this merits further investigation that could be done using frameworks like the one we describe in this paper. For example, if an investigation on the relationship between environment and song variation is suspected within species, automatically segmenting syllables from the entire range of a bird would allow researchers to test those hypotheses more readily.

## Considerations for future research

The approach we apply here is fast, flexible, and can be run on a standard laptop, though performance will benefit from parallel processing. AI approaches such as the CNN applied here can supplement the deep avian bioacoustics literature with the breadth afforded by efficient data generation. In addition, our overall framework will likely be applicable to other machine learning algorithms: TweetyNet may be replaced by other convolutional/recurrent neural network methods with these guidelines still being relevant, though this will need further investigation. It remains to be seen whether other methods, random forests or evolving neural networks, will perform similarly [100–103]. In either case, while users with laptop (or otherwise not-compute-heavy) computers should be able to run these frameworks, we do recommend that models be run in a high compute cluster setting.

Our annotated segmentation data are publicly available (via Dryad; https://doi.org/10.5061/dryad.8pk0p2nrb). To perform almost any analysis in bioacoustics at the syllable level, these segmentations—specifically the onset and offset times—are critically important for reproducibility, as without them the syllables must be re-segmented. This re-segmentation can introduce errors if multiple individuals hand-annotate. As of this writing, the authors know of one other such dataset [55]. This is despite many other studies which have performed important

bioacoustic analysis, but do not provide the raw segmentations. Further, at present we are not aware of any centralized database for annotation files, leaving researchers dependent on the availability of these annotations uploaded by others. We recommend that until such a database exists, researchers performing bioacoustic analyses should ensure that their annotations of bioacoustic files are uploaded somewhere for future work.

Automatic methods can be used to validate and supplement manual segmentation analyses, as are traditional in avian bioacoustics. From the most basic, this approach can be used with sparser forms of data, for instance continuous recordings where the species of interest only vocalizes for a short amount of time. Segmentation with this method and downstream scripts developed here can reduce the amount of recording that needs to be searched by quickly eliminating large portions of the files. This segmentation method can also be used to standardize projects among individuals who are all working concurrently. Human bias in segmentation is a known phenomenon [104–106], but applying a single model to begin from would potentially alleviate some of those issues. Further, when working from automatically segmented data, the ability to recognize incorrect or otherwise anomalous syllables where models and manual labeling do not match would make data cleaning swifter and more reliable. Lastly, the use of these models can help assist in training younger scientists without potentially compromising data quality; with a standard template to work from and the ability for the machine learning algorithm to partially check segmentations, students could spend more time developing other research skills than simple song annotation.

Automatic segmentation can also be used to set up downstream applications. While not implemented here, TweetyNet can identify individual types of syllables; indeed, this is Tweety-Net's original goal, and it does so with high accuracy. However, this requires knowledge of what types of syllables exist in the dataset, and in taxa that have not been studied before, it is hard to know *a priori* what those sounds are. It is beyond the scope of this manuscript to discuss clustering methods but being able to characterize the repertoire of a new species automatically, partially or fully, is within the realm of possibility. Further, any methods that can segment the syllables as well as classify types of syllables allow for the fine detection variation within those syllable classes, as well as changes in the syntax of bird song (i.e., the arrangement of sounds into specific sequences). This contrasts with changes in other features of the song, like frequency, duration, or bandwidth, which are independent of the order of specific syllables. The ability to categorize these sounds will be particularly useful in getting a more thorough assessment of song complexity within and between taxa.

Segmentation of sound is a critical part of analysis of song in birds, as well as other taxa that communicate with sound including some insects, frogs, and bats. Given the sheer number of species that this encompasses, having a standardized and automatic approach will allow for fast performance of traditional bioacoustic analysis. Our investigation forms a starting point for this work. Given enough time, we foresee that these and other models can be refined and used to segment all recordings available for all species, which would likely comprise multiple lifetimes of work if done manually and be intractable with the current speed at which new recordings are added. Further, as machine learning methods become more sophisticated, software could be developed to auto-segment sounds as they are cataloged into these databases and repositories, or even to be integrated into the recorders and microphones that are commonly used today.

## Conclusions

In this paper, we investigated the influence of phylogeny on machine learning methods to extract syllables from recordings of bird song, finding that the accuracy of these methods

depends in part on the phylogenetic relatedness of the taxa being trained on and tested on. However, the loss in accuracy associated with distantly related taxa can be ameliorated by including multiple species. Further, species with smaller sample sizes can be segmented with high accuracy by tuning previously trained models. We suggest the following best practices for using machine learning in bioacoustics, depending on the sample sizes of species of interest: for very small amounts of total data, it is best to hand-annotate syllables to segment them irrespective of the number of species. With larger amounts of data, hand-annotation of select species that span the phylogenetic (and bioacoustic) variation in the dataset, then training a machine learning model to segment syllables from it will reduce computational load without a major disruption to accuracy. This framework will be broadly applicable to many regions of the world and many taxa, allowing us to achieve a global perspective on vocalizations. Automated processing will make it easier to connect bioacoustics to ecological and phylogenetic studies, such that researchers who are not trained in bioacoustics can use sound data. Ultimately, this innovation will change how biologists analyze songs by expanding the scope of what is possible from depth to breadth.

## Supporting information

**S1 File.**
(PDF)

## Acknowledgments

We thank all the individuals throughout the years who have contributed to the bioacoustics repositories, including the Borror Laboratory of Bioacoustics and Xeno-Canto. For helpful feedback on these analyses, we thank D Kelly, T Berger-Wolf, G Carter, R Adams, and S Gaunt. Thoughtful reviews come from T Yuri, S Decker, F Mol Lanna, D Parsons, J Wieringa, D Duckett, S Moshier, EM Da Fonesca, MTC Thome, D Nicholson, M Weldy, and one anonymous reviewer.

## Author Contributions

**Conceptualization:** Kaiya L. Provost, Bryan C. Carstens.

**Data curation:** Kaiya L. Provost, Jiaying Yang.

**Formal analysis:** Kaiya L. Provost, Jiaying Yang.

**Funding acquisition:** Kaiya L. Provost, Bryan C. Carstens.

**Investigation:** Kaiya L. Provost, Bryan C. Carstens.

**Methodology:** Kaiya L. Provost, Bryan C. Carstens.

**Project administration:** Bryan C. Carstens.

**Resources:** Jiaying Yang, Bryan C. Carstens.

**Software:** Kaiya L. Provost.

**Supervision:** Bryan C. Carstens.

**Validation:** Kaiya L. Provost.

**Visualization:** Kaiya L. Provost.

**Writing – original draft:** Kaiya L. Provost.

**Writing – review & editing:** Kaiya L. Provost, Jiaying Yang, Bryan C. Carstens.

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
