## [Decision Letter · Decision Letter 0]

24 Jun 2022

PONE-D-22-07358The impacts of transfer learning, phylogenetic distance, and sample size on big-data bioacousticsPLOS ONE

Dear Dr. Provost,

Thank you for submitting your manuscript to PLOS ONE. After careful consideration, we feel that it has merit but does not fully meet PLOS ONE’s publication criteria as it currently stands. Therefore, we invite you to submit a revised version of the manuscript that addresses the points raised during the review process.

We look forward to receiving your revised manuscript.

Kind regards,

Anne Elizabeth Staples

Academic Editor

PLOS ONE

Journal Requirements:

“KLP and BCC were supported by NSF DEB-2016189.”

 “KLP and BCC were supported by the National Science Foundation DEB-2016189 [https://www.nsf.gov/awardsearch/showAward?AWD_ID=2016189]. The funders had no role in study design, data collection and analysis, decision to publish, or preparation of the manuscript.”

5. Please include captions for your Supporting Information files at the end of your manuscript, and update any in-text citations to match accordingly. Please see our Supporting Information guidelines for more information: http://journals.plos.org/plosone/s/supporting-information

Reviewers' comments:

Reviewer's Responses to Questions

**Comments to the Author**

1. Is the manuscript technically sound, and do the data support the conclusions?

Reviewer #1: Yes

Reviewer #2: Partly

2. Has the statistical analysis been performed appropriately and rigorously? 

Reviewer #1: Yes

Reviewer #2: Yes

3. Have the authors made all data underlying the findings in their manuscript fully available?

Reviewer #1: Yes

Reviewer #2: Yes

4. Is the manuscript presented in an intelligible fashion and written in standard English?

Reviewer #1: Yes

Reviewer #2: Yes

5. Review Comments to the Author

Reviewer #1: This paper analyzed the impact of various factors including transfer learning, phylogenetic distance and sample size on classifying the birds with their vocalizations and machine learning techniques.

Major comments:

1. This paper provided detailed explanation about data preparation and the model training setup, but the major concern is that it is not clear what questions this paper is trying to answers, or gaps to fill or how to generalize the learning from this paper. This paper started with introduction of the researches of vocalizations in birds and then the introduction (and some discussion) of machine learning models, like classic machine learning vs deep learning vs transfer learning. However, it did not mention why there is such a need. It is not clear where the research gap is and how the learning/discussion in this paper can be generalized. For example, if this paper aims to improve the state of art accuracy, it should be clearly stated in the introduction and also cite the existing works. Or if this paper aims to guide researchers in the bioacoustics area to fine tune the existing ML models, the authors should clearly state and clarify why the researcher in this area need such guideline

2. The Introduction section introduced multiple concepts: vocalization, data richness, heavy human labelling work, machine learning, deep learning, transfer learning, but they are unconnected dots and did not lead to a clear research questions. It is also not clear about the reason to introduce each topic/concept.

3. If TweetyNet is the main model to be tuned in this paper, the authors should introduce this model more and discuss about its performance, strength and weakness.

4. The annotations/labels accuracy and consistency should be measured when they are used as label to train model.

5. The Song data summary should be listed in the Method instead of Results since this is not the main purpose/result of this paper. It can also help audience to understand the data size while reading the method

6. This paper selected 6 species where the data is richly available to test the impact, please clarify why the learning from the 6 species can be generalized to all the species

7. It is commonly known that balanced dataset should have a better performance than unbalanced dataset, please clarify why the authors may think the unbalanced dataset may be better?

8. Table 2 is the accuracy of the models, it looks like a recall rate (i.e., correct prediction / ground truth). Please also provide the precision (correct prediction / all predictions for each species) for a more comprehensive assessment of the model.

In-line comments;

1. Line 69, “where the features used do not need to be pre-trained”. The word “pre-trained” may not be accurate, the word “prepared” could be better

2. Line 96, in the transfer learning, the pre-trained ANNs should not be directly applied to new, related data. Some additional training should be needed

3. Line 112, the Github link does not work

4. Line 152-155, this information should be in Introduction or mentioned at the beginning of the Method

5. Line 180-184, please provide references or rational for the parameter selection

6. Line 194, which is the reason for choosing 5-step and 50-step checkpoints? Would more checkpoints setting provide a better understanding about model performance change with the number of steps change?

Reviewer #2: This paper explored the consequences of model training decisions on the accuracy of an avian classification algorithm. The authors adopted a recently published neural network architecture which links convolutional and recurrent neural networks to predict both time-specific song segmentation boundaries (onset time and call end time) and song labels (colloquially predicting the species that made a specific call) and used this model architecture with an unbalanced (intentionally) constructed avian audio dataset to evaluate the effects of model training decisions on classification accuracy. The primary contribution of this manuscript is not the development of a new avian audio classification algorithm, but to provide practical model training advice and attempt to estimate some of the factors influencing the outcomes of model training decisions. There are substantial knowledge gaps in this field. Specifically, advances in structural or conceptual model development are far outpacing studies evaluating best practices and exploration of the factors influencing the ‘predictability’ of certain call types.

1. The study presents the results of original research.

This study presents the results of original research, and contributes to a substantial and growing knowledge gap. Advances in computer vision and hearing algorithms (and the overlap in between) and their application to ecological problems are far outpacing rigorous studies establishing best practices and evaluating the consequences of training decisions.

2. Results reported have not been published elsewhere.

To the extent of my knowledge the work presented here has not been published elsewhere. The model architecture used as a case study has been published, but the training evaluations and call complexity analyses have not been reported elsewhere.

3. Experiments, statistics, and other analyses are performed to a high technical standard and are described in sufficient detail.

The analytical framework and techniques used by the authors was reasonable. However, I have a few concerns.

A) Throughout the manuscript the authors refer to evaluations of accuracy; however, it is only inferred through context that they are only referring to classification accuracy. The TweetyNet algorithm makes two types of predictions (i.e. similar to YOLO or similar models which predict bounding boxes and classes) and both predictions can be evaluated for accuracy. The authors appear primarily concerned with an assessment of clip-level classification accuracy, which on its own makes a valuable contribution. Please make this decision explicit throughout the manuscript. But, an evaluation of temporal segmentation accuracy would extend the influence and usefulness of this evaluation in regards to establishing semi-automated labelling techniques.

B) I understand that the number of samples used for some of the species was an attempt to explore the consequences of sparsely represented classes. But, for all but two of the species, the sample sizes are well below those that would be used to develop a classification algorithm for applied projects.

C) I have some concern that there is a slight mismatch in the use of the term ‘transfer learning.’ Technically, the authors approach is a transfer learning approach (using the weights of pretrained models. However, there is a big mismatch between the general usage of the term and the application presented here. For example, transfer learning approaches are often used to harness the power of large-scale models trained across generalized datasets, which are computationally inefficient to re-run or require impractical amounts of computing power. The general thought is that the weights learned in these large-scale models are able to capture general patterns in visual or audio data and that by fine tuning small layers on top of these base weights it will be easier and faster to develop specific solutions. I have concerns that the findings presented here would not generalize to broader applications of transfer learning.

D) The authors performed a principal components analysis on the output of the SoundShape pipeline, which I assume was performed to reduce the dimensionality of the R package output. The authors then stated that the ‘used the three PCs to calculate a hypervolume per species.’ In general, PCA can be sensitive to arbitrary analytical decisions and I would feel more comfortable interpreting the results if I could see the scree-plot of the Eigenvectors or if the authors had clearly stated a selection rule. As it currently stands, this component of the analysis is not reproducible. Some other small points associated with this are that the output of a PCA can differ between packages (often due to differences in how the packages handle missing values) and the authors did not document the package, and that the irregularities presented in the results section (lines 388- 397) might be explained by the cutoff used in selecting the number of PCs to retain, particularly if the variation described by lower-level Eigenvectors was only present in the data rich species.

E) I found the hypervolume and song complexity analysis useful in thinking about the differences in classification performance; however, I feel that careful selection of study species could have increased the utility of this analysis. For example, if the authors had selected more species from the song complexity analysis [68] so they could have estimated correlation coefficients, or recreated the analysis performed in [68] so they could estimate correlations using the data on hand. I understand that the authors selected the 6 species used here because, ‘they were distributed across the phylogenetic tree and had relatively high numbers of recordings available,’ which had implications for the phylogenetic similarity analysis. However, the two conditions described above could have been satisfied by a large number of species, including a number of other species included in [68].

F) The regression analysis described on lines 252-266 is not reproducible in its current format, nor do the authors provide enough output to assess the analysis. For example, the authors state they used generalized linear regressions, ANOVA, and Tukey’s HSD tests. However, the following paragraph only discusses linear regressions (not GLMs), it is unclear if the authors fit independent univariate models or a single multivariate model (if the latter are the predictors correlated?), and there is not enough information presented to determine if the Tukey’s HSD test was employed across all comparisons as a multiple test correction or if a multiple correction was used. Lastly the regression coefficients are not presented in the analysis.

G) Briefly, some of the accuracy comparisons would have benefitted by developing bootstrapped confidence intervals. Doing so would provide more information to interpret if differences in accuracy were real or reasonable stochastic variability.

4. Conclusions are presented in an appropriate fashion and are supported by the data.

Overall, the conclusions are appropriate and there are a couple key points which make valuable contributions to the field. In particular, I found the recommendations on lines 463-469 pragmatic and useful.

I do think the paper would be stronger with some expanded discussion about the analyses and their applications. For example, there was very limited discussion about the limitations of the call song complexity and hypervolume analysis or the ecological or phylogenetic implications of this analysis. Perhaps the limited scope of this analytical component could be effectively supplemented with a visual representation of the different songs. I would expect calls with similar hypervolumes to appear visually similar (spanning long frequency or temporal windows).

I would prefer additional detail in some sections, but to some extent this is stylistic. I do caution the authors to keep the discussion and conclusion sections focused on their contributions and not on the contributions made by the authors of the TweetyNet algorithm. For example, lines 488-490 claim that the authors “application can supplement the deep avian bioacoustics literature with the breadth afforded by efficient data generation.” I would argue that these contributions and the application were made by the authors of TweetyNet and that the authors of this manuscript did not develop an application or new processing algorithm that reduces the burden of efficient data generation (i.e. labeling). In addition, I felt the discussion about clustering-based classification (lines 514-518) outside the scope of the analysis.

5. The article is presented in an intelligible fashion and is written in standard English.

The manuscript is presented in standard English. There are a number of confusing run-on sentences, including a paragraph consisting of a single 4-clause sentence near the end of the introduction. In addition, there are a few places where I feel (and in full disclosure that this is opinion, so the authors can address as they see fit) the authors chose concise or loose sentence structures at the expense of clarity. For example, on lines 291-292 it is unclear that the highlighted findings are in reference to the transfer learning conditions.

Broadly speaking, a careful attempt to reduce jargon throughout the paper and clearly define terms when they are standard or necessary (“zero-shot”, “few-shot”) would make the paper more accessible.

One specific spot I would recommend revisiting would be the first paragraph of the Machine Learning section in the Introduction. In particular the second half of the paragraph was not informative (e.g. what is a slew of problems, why is that slew important, are there specific tensions between these problems and ecological data types). Also, at least 2 of the three machine learning algorithms listed have deep roots in probabilistic traditions (linear regression, and MaxEnt (as a Poisson process). Although, both have been adopted to ‘learning’ inferential paradigms.

6. The research meets all applicable standards for the ethics of experimentation and research integrity.

This paper does not violate any ethics of experimentation. However, I have some concerns from an analytical perspective that the authors performed a number of statistical tests (multiple regressions), did not use a multiple test correction (Bonferroni etc…), and only presented a p-value based summary for some of the tests instead of magnitudes and confidence intervals.

7. The article adheres to appropriate reporting guidelines and community standards for data availability.

The authors claim that the data is available without restrictions. I was able to verify access to some of the files on the Borror Lab of Bioacoustics database and Xeno-Canto (I did not test access to all the files). The code repository is not currently accessible so I am unable to verify.

6. PLOS authors have the option to publish the peer review history of their article (what does this mean?). If published, this will include your full peer review and any attached files.

Reviewer #1: No

Reviewer #2: **Yes: **Matthew Weldy

---

## [Author Response · Author response to Decision Letter 0]

19 Oct 2022

Thank you very much for all of your comments. In this letter we have set off our responses in a different font as well as color, and they are also marked with numbers before each response. Please also find attached a PDF. 

PONE-D-22-07358

The impacts of transfer learning, phylogenetic distance, and sample size on big-data bioacoustics

PLOS ONE

Dear Dr. Provost,

Thank you for submitting your manuscript to PLOS ONE. After careful consideration, we feel that it has merit but does not fully meet PLOS ONE’s publication criteria as it currently stands. Therefore, we invite you to submit a revised version of the manuscript that addresses the points raised during the review process.

“KLP and BCC were supported by NSF DEB-2016189.”

 “KLP and BCC were supported by the National Science Foundation DEB-2016189 [https://www.nsf.gov/awardsearch/showAward?AWD_ID=2016189]. The funders had no role in study design, data collection and analysis, decision to publish, or preparation of the manuscript.”

5. Please include captions for your Supporting Information files at the end of your manuscript, and update any in-text citations to match accordingly. Please see our Supporting Information guidelines for more information: http://journals.plos.org/plosone/s/supporting-information

Reviewers' comments:

Reviewer #1: This paper analyzed the impact of various factors including transfer learning, phylogenetic distance and sample size on classifying the birds with their vocalizations and machine learning techniques.

Thank you very much for your comments!

Major comments:

1. This paper provided detailed explanation about data preparation and the model training setup, but the major concern is that it is not clear what questions this paper is trying to answers, or gaps to fill or how to generalize the learning from this paper. This paper started with introduction of the researches of vocalizations in birds and then the introduction (and some discussion) of machine learning models, like classic machine learning vs deep learning vs transfer learning. However, it did not mention why there is such a need. It is not clear where the research gap is and how the learning/discussion in this paper can be generalized. For example, if this paper aims to improve the state of art accuracy, it should be clearly stated in the introduction and also cite the existing works. Or if this paper aims to guide researchers in the bioacoustics area to fine tune the existing ML models, the authors should clearly state and clarify why the researcher in this area need such guideline

Thank you for pointing out this clear omission. We worked to clarify the text in the entire manuscript to make it clearer both to experts and non-experts why this work is needed and how it points the way to future research.

2. The Introduction section introduced multiple concepts: vocalization, data richness, heavy human labeling work, machine learning, deep learning, transfer learning, but they are unconnected dots and did not lead to clear research questions. It is also not clear about the reason to introduce each topic/concept.

We have clarified this and added better transition sentences between the sections. 

3. If TweetyNet is the main model to be tuned in this paper, the authors should introduce this model more and discuss about its performance, strength and weakness.

We have discussed pros and cons of using TweetyNet. Some of this discussion is in the TweetyNet paper itself and so we summarize that. 

4. The annotations/labels accuracy and consistency should be measured when they are used as label to train model.

We have added more information about model performance to address this concern, including accuracy, recall, precision, and F-score measures as well as segment error rate.

5. The Song data summary should be listed in the Method instead of Results since this is not the main purpose/result of this paper. It can also help audience to understand the data size while reading the method

We have moved this section into the methods. 

6. This paper selected 6 species where the data is richly available to test the impact, please clarify why the learning from the 6 species can be generalized to all the species

We added more species that were likewise distributed across the tree. These species do not have as many recordings available as the original selection but they overlap more closely with the complexity dataset. We now have a total of 9 species, where all but 3 do not have data from the Medina and Francis 2012 manuscript. We have also shown that the correlation between complexity and performance is still there. We feel confident to say that in Passeriform birds (~70% of extant species) our method will work. 

7. It is commonly known that balanced dataset should have a better performance than unbalanced dataset, please clarify why the authors may think the unbalanced dataset may be better?

We have omitted the unbalanced dataset from the final manuscript. 

8. Table 2 is the accuracy of the models, it looks like a recall rate (i.e., correct prediction / ground truth). Please also provide the precision (correct prediction / all predictions for each species) for a more comprehensive assessment of the model.

We have calculated precision, recall, and f-score for each of the models. 

In-line comments;

1. Line 69, “where the features used do not need to be pre-trained”. The word “pre-trained” may not be accurate, the word “prepared” could be better

We have changed this.

2. Line 96, in the transfer learning, the pre-trained ANNs should not be directly applied to new, related data. Some additional training should be needed

We realized after these revisions that we were using “transfer learning” incorrectly in this context. What we want to know is whether models trained on one species can be applied to another without additional training. We have updated the text to reflect this. 

3. Line 112, the Github link does not work

We had accidentally set the GitHub link to be a private repository. We have fixed this.

4. Line 152-155, this information should be in Introduction or mentioned at the beginning of the Method

Fixed

5. Line 180-184, please provide references or rational for the parameter selection

Fixed

6. Line 194, which is the reason for choosing 5-step and 50-step checkpoints? Would more checkpoints setting provide a better understanding about model performance change with the number of steps change?

We have discussed this method with the TweetyNet developers who instead informed us we needed to proceed for longer checkpoints (100) and also change our parameter space. Originally we wanted to include this to verify whether training models for longer would cause them to perform better -- it does. However we do not think that this is relevant anymore and as such we have omitted all comparisons between checkpoints. 

Reviewer #2: This paper explored the consequences of model training decisions on the accuracy of an avian classification algorithm. The authors adopted a recently published neural network architecture which links convolutional and recurrent neural networks to predict both time-specific song segmentation boundaries (onset time and call end time) and song labels (colloquially predicting the species that made a specific call) and used this model architecture with an unbalanced (intentionally) constructed avian audio dataset to evaluate the effects of model training decisions on classification accuracy. The primary contribution of this manuscript is not the development of a new avian audio classification algorithm, but to provide practical model training advice and attempt to estimate some of the factors influencing the outcomes of model training decisions. There are substantial knowledge gaps in this field. Specifically, advances in structural or conceptual model development are far outpacing studies evaluating best practices and exploration of the factors influencing the ‘predictability’ of certain call types.

Thank you Matthew Weldy for your comments!

1. The study presents the results of original research.

This study presents the results of original research, and contributes to a substantial and growing knowledge gap. Advances in computer vision and hearing algorithms (and the overlap in between) and their application to ecological problems are far outpacing rigorous studies establishing best practices and evaluating the consequences of training decisions.

2. Results reported have not been published elsewhere.

To the extent of my knowledge the work presented here has not been published elsewhere. The model architecture used as a case study has been published, but the training evaluations and call complexity analyses have not been reported elsewhere.

3. Experiments, statistics, and other analyses are performed to a high technical standard and are described in sufficient detail.

The analytical framework and techniques used by the authors was reasonable. However, I have a few concerns.

A) Throughout the manuscript the authors refer to evaluations of accuracy; however, it is only inferred through context that they are only referring to classification accuracy. The TweetyNet algorithm makes two types of predictions (i.e. similar to YOLO or similar models which predict bounding boxes and classes) and both predictions can be evaluated for accuracy. The authors appear primarily concerned with an assessment of clip-level classification accuracy, which on its own makes a valuable contribution. Please make this decision explicit throughout the manuscript. But, an evaluation of temporal segmentation accuracy would extend the influence and usefulness of this evaluation in regards to establishing semi-automated labeling techniques.

We were specifically looking at clip level classification accuracy, yes, but we have also added syllable error rate which is an assessment of segmentation accuracy into the analysis. 

B) I understand that the number of samples used for some of the species was an attempt to explore the consequences of sparsely represented classes. But, for all but two of the species, the sample sizes are well below those that would be used to develop a classification algorithm for applied projects.

Unfortunately, this reflects the reality of the data we are seeking to train on more broadly. Most bird species do not have sufficient sample sizes to classify their syllables alone. However, we are not looking to use a classification algorithm that identifies the species -- rather, we are trying to identify the presence (as well as onset and offset) of syllables. All birds that vocalize have syllables although they are variable between species. Further, we chose TweetyNet in particular because its underlying cNN+rNN framework was shown to be highly accurate with these limited datasets. Our goal for this manuscript is to create use-cases for later researchers such that they do not spend their precious research time parameterizing models and annotating data. 

C) I have some concern that there is a slight mismatch in the use of the term ‘transfer learning.’ Technically, the authors approach is a transfer learning approach (using the weights of pretrained models. However, there is a big mismatch between the general usage of the term and the application presented here. For example, transfer learning approaches are often used to harness the power of large-scale models trained across generalized datasets, which are computationally inefficient to re-run or require impractical amounts of computing power. The general thought is that the weights learned in these large-scale models are able to capture general patterns in visual or audio data and that by fine tuning small layers on top of these base weights it will be easier and faster to develop specific solutions. I have concerns that the findings presented here would not generalize to broader applications of transfer learning.

Thank you for this comment. Biological data are unique in the sense that there is shared history across the units of analysis. In this case, we analyze data from different species, yet these data are not statistically independent due to common ancestry (i.e., their phylogenetic history). For this reason it isn’t possible to create a ‘generalized’ data set of bird song recordings because there are hidden correlations among the individual calls due to this ancestry. We anticipate that these findings (i.e., particularly that the utility of a model trained using data from species ‘A’ would be applicable to species ‘B’ would have broad application to biological data in a manner that is proportional to their phylogenetic distance), but of course have not conducted this analysis. 

D) The authors performed a principal components analysis on the output of the SoundShape pipeline, which I assume was performed to reduce the dimensionality of the R package output. The authors then stated that the ‘used the three PCs to calculate a hypervolume per species.’ In general, PCA can be sensitive to arbitrary analytical decisions and I would feel more comfortable interpreting the results if I could see the scree-plot of the Eigenvectors or if the authors had clearly stated a selection rule. As it currently stands, this component of the analysis is not reproducible. Some other small points associated with this are that the output of a PCA can differ between packages (often due to differences in how the packages handle missing values) and the authors did not document the package, and that the irregularities presented in the results section (lines 388- 397) might be explained by the cutoff used in selecting the number of PCs to retain, particularly if the variation described by lower-level Eigenvectors was only present in the data rich species.

This is a good critique. To address this we set a cutoff of 50% accuracy. We originally were doing this with the first 3 PCs due to memory constraints with the convex hull algorithm for calculating hypervolumes. However we have changed our analysis to alleviate those constraints. 

E) I found the hypervolume and song complexity analysis useful in thinking about the differences in classification performance; however, I feel that careful selection of study species could have increased the utility of this analysis. For example, if the authors had selected more species from the song complexity analysis [68] so they could have estimated correlation coefficients, or recreated the analysis performed in [68] so they could estimate correlations using the data on hand. I understand that the authors selected the 6 species used here because, ‘they were distributed across the phylogenetic tree and had relatively high numbers of recordings available,’ which had implications for the phylogenetic similarity analysis. However, the two conditions described above could have been satisfied by a large number of species, including a number of other species included in [68].

We added more species from this complexity analysis to our dataset here. Our results do not change appreciably -- more complex species are harder to optimize. 

F) The regression analysis described on lines 252-266 is not reproducible in its current format, nor do the authors provide enough output to assess the analysis. For example, the authors state they used generalized linear regressions, ANOVA, and Tukey’s HSD tests. However, the following paragraph only discusses linear regressions (not GLMs), it is unclear if the authors fit independent univariate models or a single multivariate model (if the latter are the predictors correlated?), and there is not enough information presented to determine if the Tukey’s HSD test was employed across all comparisons as a multiple test correction or if a multiple correction was used. Lastly the regression coefficients are not presented in the analysis.

We clarified this in the text. Further we have put a full list of the models used and all of their associated coefficients into the supplement. 

G) Briefly, some of the accuracy comparisons would have benefitted by developing bootstrapped confidence intervals. Doing so would provide more information to interpret if differences in accuracy were real or reasonable stochastic variability.

We now also report mean and standard deviations of accuracy and other performance metrics along with our point estimates. 

4. Conclusions are presented in an appropriate fashion and are supported by the data.

Overall, the conclusions are appropriate and there are a couple key points which make valuable contributions to the field. In particular, I found the recommendations on lines 463-469 pragmatic and useful.

Thank you!

I do think the paper would be stronger with some expanded discussion about the analyses and their applications. For example, there was very limited discussion about the limitations of the call song complexity and hypervolume analysis or the ecological or phylogenetic implications of this analysis. Perhaps the limited scope of this analytical component could be effectively supplemented with a visual representation of the different songs. I would expect calls with similar hypervolumes to appear visually similar (spanning long frequency or temporal windows).

As you can see in the sonograms in Figure 2, this is true in some cases but not all. We have improved discussion of this figure in an effort to better make this point. 

I would prefer additional detail in some sections, but to some extent this is stylistic. I do caution the authors to keep the discussion and conclusion sections focused on their contributions and not on the contributions made by the authors of the TweetyNet algorithm. For example, lines 488-490 claim that the authors “application can supplement the deep avian bioacoustics literature with the breadth afforded by efficient data generation.” I would argue that these contributions and the application were made by the authors of TweetyNet and that the authors of this manuscript did not develop an application or new processing algorithm that reduces the burden of efficient data generation (i.e. labeling). 

Thank you for this comment. We have attempted to focus our discussion on the contributions of this work rather than also including aspects of TweetyNet. 

In addition, I felt the discussion about clustering-based classification (lines 514-518) outside the scope of the analysis.

This has been removed. Clustering in this case is particularly important when moving from segmentation to syntax, but that’s beyond the scope of this manuscript at present. 

5. The article is presented in an intelligible fashion and is written in standard English.

The manuscript is presented in standard English. There are a number of confusing run-on sentences, including a paragraph consisting of a single 4-clause sentence near the end of the introduction. In addition, there are a few places where I feel (and in full disclosure that this is opinion, so the authors can address as they see fit) the authors chose concise or loose sentence structures at the expense of clarity. For example, on lines 291-292 it is unclear that the highlighted findings are in reference to the transfer learning conditions.

Broadly speaking, a careful attempt to reduce jargon throughout the paper and clearly define terms when they are standard or necessary (“zero-shot”, “few-shot”) would make the paper more accessible.

One specific spot I would recommend revisiting would be the first paragraph of the Machine Learning section in the Introduction. In particular the second half of the paragraph was not informative (e.g. what is a slew of problems, why is that slew important, are there specific tensions between these problems and ecological data types). Also, at least 2 of the three machine learning algorithms listed have deep roots in probabilistic traditions (linear regression, and MaxEnt (as a Poisson process). Although, both have been adopted to ‘learning’ inferential paradigms.

Thank you for highlighting this. We have gone through the manuscript carefully with an eye towards improving readability and revising the writing to be more concise and direct. 

6. The research meets all applicable standards for the ethics of experimentation and research integrity.

This paper does not violate any ethics of experimentation. However, I have some concerns from an analytical perspective that the authors performed a number of statistical tests (multiple regressions), did not use a multiple test correction (Bonferroni etc…), and only presented a p-value based summary for some of the tests instead of magnitudes and confidence intervals.

We have used a Bonferroni correction -- thank you for the suggestion. 

7. The article adheres to appropriate reporting guidelines and community standards for data availability.

The authors claim that the data is available without restrictions. I was able to verify access to some of the files on the Borror Lab of Bioacoustics database and Xeno-Canto (I did not test access to all the files). The code repository is not currently accessible so I am unable to verify.

We accidentally set the code repository to private instead of unlisted. It is available now.

---

## [Decision Letter · Decision Letter 1]

18 Nov 2022

The impacts of fine-tuning, phylogenetic distance, and sample size on big-data bioacoustics

PONE-D-22-07358R1

Dear Dr. Provost,

We’re pleased to inform you that your manuscript has been judged scientifically suitable for publication and will be formally accepted for publication once it meets all outstanding technical requirements.

Kind regards,

Anne Elizabeth Staples

Academic Editor

PLOS ONE

Additional Editor Comments (optional):

Please fix the following issues in your manuscript, as recommended by Reviewer 1 upon re-review of your revised manuscript:

The authors have done a laudable job revising their manuscript and addressing comments made during the first round of review. I think this manuscript should be accepted and I thank the authors for their contribution. I've enjoyed reading their work and considering a new (to me) bioacoustic perspective.

Below are a few minor comments for the authors.

Lines 70-72. Consider merging this sentence with the last paragraph of the Introduction.

Lines 139-147. After re reading the section describing the linear models fit to assess model performance and fit time relative to measures of call complexity and taxonomic divergence times, it is clear that the authors gave this quite a bit of thought and had some fairly well developed hypotheses. I think the manuscript would benefit from trying to lay out some of those hypotheses here . Not in the sense of describing each relationship among species, but at the level of we hypothesized that as species divergence time increased the models would transfer less successfully.

Lines 151-152. I am still unable to access the github repository.

Line 346. If the relationship between the response and the linear model is not mediated by a link function (other than the identity) than these model fits should be referred to as just linear models. The generalized component is an indicator that a link function was used or that the residual error of the model is non-normally distributed.

Line 461. Change 'through time' to 'as the training set size increased.'

Lines 507-510. My interpretation is that the error values indicated in parentheses here should line up with the mean values from the SER column of Table 2. If that is correct, these values are incorrect. If I am mistaken, consider expanding the description a little for clarity.

Reviewers' comments:

Reviewer's Responses to Questions

**Comments to the Author**

1. If the authors have adequately addressed your comments raised in a previous round of review and you feel that this manuscript is now acceptable for publication, you may indicate that here to bypass the “Comments to the Author” section, enter your conflict of interest statement in the “Confidential to Editor” section, and submit your "Accept" recommendation.

Reviewer #2: All comments have been addressed

2. Is the manuscript technically sound, and do the data support the conclusions?

Reviewer #2: Yes

3. Has the statistical analysis been performed appropriately and rigorously? 

Reviewer #2: Yes

4. Have the authors made all data underlying the findings in their manuscript fully available?

Reviewer #2: Yes

5. Is the manuscript presented in an intelligible fashion and written in standard English?

Reviewer #2: Yes

6. Review Comments to the Author

Reviewer #2: The authors have done a laudable job revising their manuscript and addressing comments made during the first round of review. I think this manuscript should be accepted and I thank the authors for their contribution. I've enjoyed reading their work and considering a new (to me) bioacoustic perspective.

Below are a few minor comments for the authors.

Lines 70-72. Consider merging this sentence with the last paragraph of the Introduction.

Lines 139-147. After re reading the section describing the linear models fit to assess model performance and fit time relative to measures of call complexity and taxonomic divergence times, it is clear that the authors gave this quite a bit of thought and had some fairly well developed hypotheses. I think the manuscript would benefit from trying to lay out some of those hypotheses here . Not in the sense of describing each relationship among species, but at the level of we hypothesized that as species divergence time increased the models would transfer less successfully.

Lines 151-152. I am still unable to access the github repository.

Line 346. If the relationship between the response and the linear model is not mediated by a link function (other than the identity) than these model fits should be referred to as just linear models. The generalized component is an indicator that a link function was used or that the residual error of the model is non-normally distributed.

Line 461. Change 'through time' to 'as the training set size increased.'

Lines 507-510. My interpretation is that the error values indicated in parentheses here should line up with the mean values from the SER column of Table 2. If that is correct, these values are incorrect. If I am mistaken, consider expanding the description a little for clarity.

7. PLOS authors have the option to publish the peer review history of their article (what does this mean?). If published, this will include your full peer review and any attached files.

Reviewer #2: **Yes: **Matthew Weldy

---

## [Editor Report · Acceptance letter]

29 Nov 2022

PONE-D-22-07358R1 

The impacts of fine-tuning, phylogenetic distance, and sample size on big-data bioacoustics 

Dear Dr. Provost:

I'm pleased to inform you that your manuscript has been deemed suitable for publication in PLOS ONE. Congratulations! Your manuscript is now with our production department. 

Kind regards, 

on behalf of

Dr. Anne Elizabeth Staples 

Academic Editor

PLOS ONE